

# Syntactic model-based human body 3D reconstruction and event classification via association based features mining and deep learning

Yazeed Ghadi[1], Israr Akhter[2], Mohammed Alarfaj[3], Ahmad Jalal[2] and Kibum Kim[4]

[1] Department of Computer Science and Software Engineering, Al Ain University, Al Ain, UAE
[2] Department of Computer Science, Air University, Islamabad, Pakistan
[3] Department of Electrical Engineering, College of Engineering, King Faisal University, Al-Ahsa, Saudi Arabia
[4] Department of Human-Computer Interaction, Hanyang University, Ansan, South Korea

Corresponding author
Kibum Kim, kibum@hanyang.ac.kr

## ABSTRACT

The study of human posture analysis and gait event detection from various types of inputs is a key contribution to the human life log. With the help of this research and technologies humans can save costs in terms of time and utility resources. In this paper we present a robust approach to human posture analysis and gait event detection from complex video-based data. For this, initially posture information, landmark information are extracted, and human 2D skeleton mesh are extracted, using this information set we reconstruct the human 2D to 3D model. Contextual features, namely, degrees of freedom over detected body parts, joint angle information, periodic and non-periodic motion, and human motion direction flow, are extracted. For features mining, we applied the rule-based features mining technique and, for gait event detection and classification, the deep learning-based CNN technique is applied over the mpii-video pose, the COCO, and the pose track datasets. For the mpii-video pose dataset, we achieved a human landmark detection mean accuracy of 87.09% and a gait event recognition mean accuracy of 90.90%. For the COCO dataset, we achieved a human landmark detection mean accuracy of 87.36% and a gait event recognition mean accuracy of 89.09%. For the pose track dataset, we achieved a human landmark detection mean accuracy of 87.72% and a gait event recognition mean accuracy of 88.18%. The proposed system performance shows a significant improvement compared to existing state-of-the-art frameworks.

# INTRODUCTION

With regard to human posture information, motion estimation, and gait event detection from various types of input such as camera-based data, sensor-based datasets currently provide the most challenging issues. Various approaches and models are proposed to find more accurate and appropriate methods and functions for event classification and human body posture, motion and movement analysis. Data generation, communication, and

transmission are routine takes for smart systems such as hospital management systems, educational systems, emergency systems, communication systems (*Tahir, Jalal & Kim, 2011*; *Tahir, 2020*), store records, airport or other transportation systems (*Jalal, Akhtar & Kim, 2020*; *Jalal, Khalid & Kim, 2020*; *Khalid et al., 2021*). The generated data is needed to be processed and utilized in the context of finding some useful information for humans in terms of time-saving, and reducing the cost of manpower (*Jalal, Akhtar & Kim, 2020*; *Jalal, Khalid & Kim, 2020*). Human posture and gait event analysis can help us represent human movement as information that is useful in smart systems for the detection and identification of human events and conditions such as standing, walking, running, playing, singing and dancing.

Various smart systems (such as smart surveillance systems, cryptography in smart systems, and data security systems; *Ur Rehman, Raza & Akhter, 2018*), management systems and smart sports systems save us time and manpower and thus also money. With smart systems, many aspects of the behavior and condition of patients can be monitored automatically and more efficiently, thus relieving the burden on medical staff and patient care and freeing them from some redundancies. For sports, we can recognize the current event such as classification about games. Smart security systems can detect the security status, security issues, normal and abnormal events in places such as airports, or defense facilities.

Following are the main key points of problem statements.

- In various crowd-based systems, human motion analysis is a challenge to perform and archive.
- For human motion analysis, mostly 2D human model was suggested and, in this regard, 3D reconstruction is proposed.
- For data optimization, most of the research was based on traditional data optimization techniques. To deal with this, data mining techniques are proposed.
- Event recognition is challenging in the computer vision domain, especially when complex human crowd-based data is used as an input of the system. To tackle this problem, we adopted contextual features and CNN-based classification.

## Technical gaps

In this domain, the various model was proposed with various techniques. Many papers deal with human motion analysis and gait event detection separately. In the above papers, research gaps and technical gaps are related throughout input–output. Initially, input is based on video-based data, and pre-pressing is performed to deal with data processing cast and time-saving. After that, the motion analysis of human detection, human landmark, human 2D stick model, and 3D human reconstruction are performed. This technique is robust to find and analyze the human motion information data which was normally dealt with in previous papers and was based on the gap of human motion methods. For event recognition, the pre-step to find the features with a data mining approach using CNN gait event detection and recognition is performed. On the other hand, previous papers deal with various models and data optimization which was the issue of processing cast and time.

## Contribution of this paper

Thus, in this research article, we propose a robust method for human body posture analysis and human gait event detection. For this, we use the mpii-video-pose dataset; the COCO dataset and the pose track dataset as input, and initially preprocess the video samples for frame conversion, motion blur noise reduction and resizing. The next step in human detection is to process these data sources by various algorithms for silhouette extraction, optimization of detected human silhouettes and human body landmark detection. Then, the next stage is to analyze the human posture information. For this, given 2D images to approximate 3D image reconstruction is applied using ellipsoid and synthetic modeling of the human body. This is followed by the feature extraction phase in which contextual features information is extracted. This information includes the degree of freedom, periodic motion, non-periodic motion, motion direction, flow, and rotational and angular joint features. For features mining, an association-based technique is adopted. For gait event classification, CNN is applied.

The main contribution of this paper is:

- With complex datasets, the detection of and optimization of humans and the extraction of and optimization of human silhouettes are challenging. We, therefore, propose a robust human silhouette extraction approach.
- For human motion, posture, and movement information analysis, we propose a method for the conversion of the human skeleton-based 2D mesh to the 3D human skeleton.
- For the detection and recognition of gait events and contextual features, the extraction approaches are proposed in which degrees of freedom (DOF), periodic motion, non-periodic motion, motion direction, flow, and rotational angular joint features are extracted.
- Finally, data mining and classification *via* hierarchical methods, mining, and CNN-based methods are adopted for gait event classification.

The subdivisions of this article are as follows: we start with related works, followed by our system methodology, then, the detailed experimental setup discussion and, finally, an overview of the paper is presented in the conclusion.

## RELATED WORK

Innovations in smartphone cameras and recorded video and developments in object marker sensor-based devices allow for more efficient farming and collection of data for exploration and research in the area. Several novel and effective approaches for recognizing human events, movements, and postures have been developed in the past. Table 1 includes a comprehensive review of recent research in this area.

## MATERIALS & METHODS

For the video input of our proposed method, the main source is RGB cameras which provide clean and noise-free data. The first step is video to frame conversion 30 fps which reduces computational cost and time. After this, noise reduction techniques are applied.

**Table 1  Comprehensive review of relevant research.**

| Human 2D posture analysis and event detection | |
| --- | --- |
| **Methods** | **Main contributions** |
| *Liu, Luo & Shah (2009)* | Using contextual, stationary, and vibration attributes, an effective randomized forest-based methodology for human body part localization was developed. They used videos and photographs to evaluate different human actions. |
| *Khan et al. (2020a)* and *Khan et al. (2020b)* | A micro, horizontal, and vertical differential function was proposed as part of an automated procedure. To classify human behavior, they used Deep Neural Network (DNN) mutation. To accomplish DNN-based feature strategies, a pre-trained Convolutional Neural Network Convolution layer was used. |
| *Zou et al. (2020)* | Adaptation-Oriented Features (AOF), an integrated framework with one-shot image classification for approximation to human actions was defined. The system applies to all classes, and they incorporated AOF parameters for enhanced performance. |
| *Franco, Magnani & Maio (2020)* | They created a multilayer structure with significant human skeleton details using RGB images. They used Histogram of Oriented (HOG) descriptor attributes to identify human actions. |
| *Ullah et al. (2019)* | The defined a single Convolutional Neural Network (CNN)-based actual data communications and information channel method. They utilized vision methods to gather information through non-monitoring instruments. The Convolutional Neural Network (CNN) technique is used to predict temporal features as well as deep auto-encoders and deep features in order to monitor human behavior. |
| *Van der Kruk & Reijne (2018)* | They developed an integrated approach to calculate vibrant human motion in sports events using movement tracker sensors. The major contribution is the computation of human events in sports datasets by estimating the kinematics of human body joints, motion, velocity, and recreation of the human pose. |
| *Wang & Mori (2008)* | They developed a lightweight event recognition strategy based on spatial development and social body pose. The kinematics knowledge of attached human body parts is used to characterize tree-based characteristics. |
| *Amft & Tröster (2008)* | Using a Hidden Markov methodology, they built a solid framework for event identification which is accomplished using time-continuous dependent features and body marker detectors. |
| *Wang et al. (2019)* | With the assistance of a human tracking methodology, they developed a comprehensive new approach for estimating the accuracy of human motion. The Deep Neural Network (DNN) is used to identify events. |
| *Jiang et al. (2015)* | They introduced a multidimensional function method for estimating human motion and gestures. They used a late mean combination algorithm to recognize events in complex scenes. |

**Table 1** (*continued*)

| Human 2D posture analysis and event detection | |
| --- | --- |
| **Methods** | **Main contributions** |
| *Li et al. (2020)* | They developed a lightweight organizational approach focused on optimal allocation, optical flow, and a histogram of the extracted optical flow. They were able to achieve effective event recognition using the standard optimization process, body joint restoration, and a Reduced and Compressed Coefficient Dictionary Learning (LRCCDL) methodology. |
| *Einfalt et al. (2019)* | Through task identification, isolation of sequential 2D posture characteristics and a convolutional sequence network, a coherent framework for event recognition with athletes in motion was created. They correctly identified number of sporting event. |
| *Yu, Lei & Hu (2019)* | Their work describes a probabilistic framework for detecting events in specific interchanges in soccer rivalry videos. This is done using the replay recognition approach which recognizes the most important background features for fulfilling spectator needs and generating replay storytelling clips. |
| *Franklin, Mohana & Dabbagol (2020)* | A comprehensive deep learning framework for identifying anomalous and natural events was developed. The findings were obtained using differentiation, grouping, and graph-based techniques. They discovered natural and unusual features for event duration use using deep learning techniques. |
| *Cao et al. (2017)* | This article is based on a real-time method for detecting the 2D posture of numerous individuals in a picture. The suggested technique learns to connect body parts with persons in the image using a nonparametric informed decision-making Part Affinity Fields (PAFs). |
| **3D human posture analysis and event detection** | |
| *Aggarwal & Cai (1999)* | They devised a reliable method for analyzing the movement of human body parts through multiple cameras which monitor the body parts detection. They also created a simulation for human body joints that is 2D-3D. |
| *Hassner & Basri (2006)* | They designed an example-based synthesis methodology using a single class-based objects database that holds example reinforcements of realistic mappings due to the complexity of the objects. |
| *Hu et al. (2004)* | To define facial dimensionality, an effective 2D-to-3D hybrid face reconstruction technique is used to recreate a customizable 3D face template from a single cortical face picture with a neutral expression and regular lighting. Immersive-looking faces including different PIE are synthesized based on the customizable 3D image. |

**Table 1** (*continued*)

| Human 2D posture analysis and event detection | |
|---|---|
| **Methods** | **Main contributions** |
| *Zheng et al. (2011)* | To enhance the classification of both the roots from each 2D image, they initially model the context only as a harmonic function. Second, they analyze the formalized graphical hull definition, which eliminates jitter and diffusion by maintaining continuity with a single 2D image. Third, they maintain connectivity by making variations to the 3D reconstruction by global errors minimization. |
| *Uddin et al. (2011)* | They proposed a heuristic approach for human activity detection and human posture analysis. For this, they utilized human body joint angle information with the help of the hidden Markov model (HMM). |
| *Lohithashva, Aradhya & Guru (2020)* | The researchers created a deep learning system for detecting abnormal and normal events. Distinction, classification, and graph-based methods were used to obtain the results. Using deep learning methods, they explored natural and uncommon features for event interval use. |
| *Feng et al. (2020)* | To retrieve deep features' spatial locations in composite images, a guided Long Short-Term Memory (LSTM) approach that is based on a Convolutional Neural Network (CCN) system was evaluated. For personal authentication, the state-of-the-art YOLO v3 template was used and, for event recognition, a directed Long Short-Term Memory (LSTM) driven method was used. |
| *Khan et al. (2020a)* and *Khan et al. (2020b)* | They developed home-based patient control strategies based on body-marker detectors. To record data from patients, body-marker sensors with a color indicator framework are connected to the joints. |
| *Mokhlespour Esfahani et al. (2017)* | For sporting events, human movement monitoring body-marker tools were used to establish a Trunk Motion Method (TMM) with Body-worn Sensors that provide a low power physical system (BWS). Twelve removable detectors were used to measure 3D trunk movements in this process. |
| *Golestani & Moghaddam (2020)* | A robust wireless strategy was developed for detecting physical human behavior. They used a magnetic flux cable to monitor human behavior, and thematic maps were attached to the body joints. Research lab approximation function and Deep RNN (Recurrent Neural Network) were used to enhance efficiency. |

Human detection is achieved using Markov random field, change detection, floor detection, and spatial–temporal dereferencing. Then, 3D human reconstruction is achieved in which computational models with ellipsoids, the synthetic model with supper quadrics, joint angle estimation, and 3D reconstruction are applied. After this contextual features extraction is applied. For features mining we applied association-based techniques. Finally, gait event classification is achieved with the help of CNN over three state-of-the-art datasets. Figure 1 demonstrates the proposed system model's structural design.

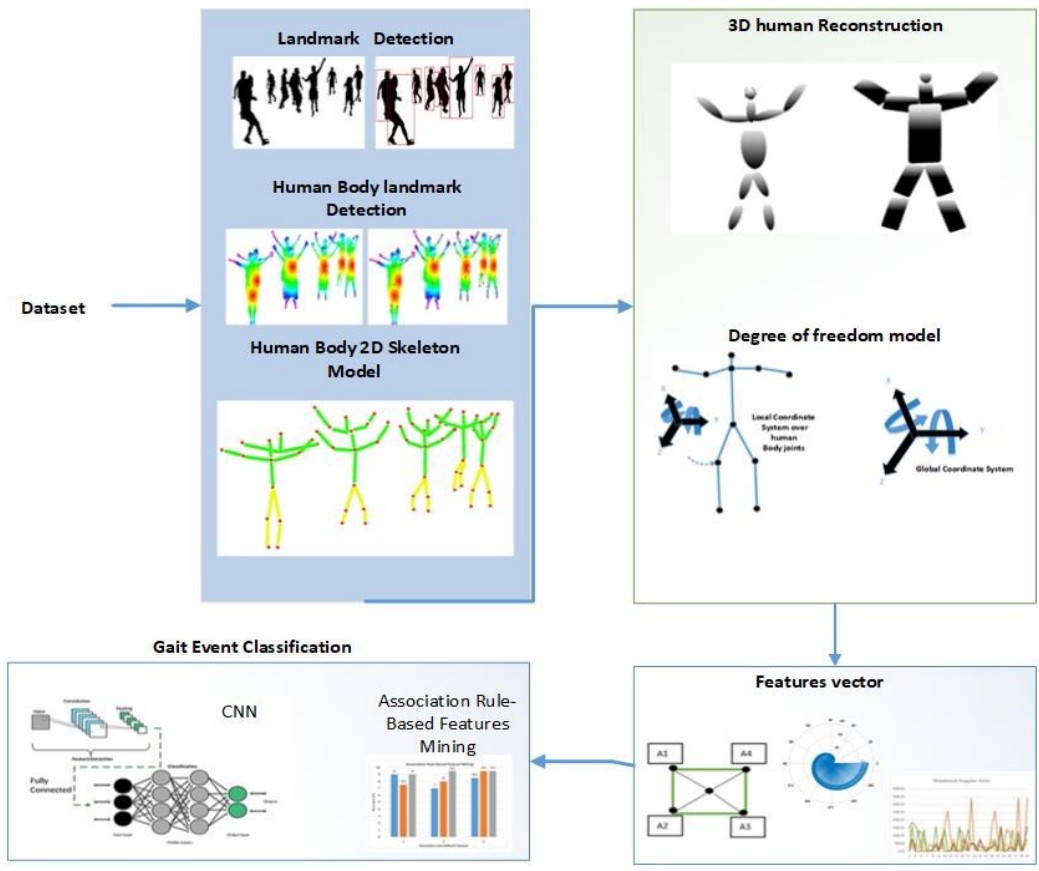

**Figure 1** The proposed system model's structural design.

    The complete overview of the proposed method is described in (Algorithm 1) in which steps of the proposed method are described and main functions with sub-functions and equations are mentions.

| Algorithm 1: Overview of the proposed method |
| --- |

**Step 01.**     **Input (Video based data set) Iv**

        Fun(Input)

        Video_data1, Video_data2, Video_data3

**Step 02.**     **Preprocessing on video data  Pd**

        Fun(Nosie  redu)

        Fun(Frame  conv)

        Fun(Gry_scale, Bin_Covn)

     a.    Noise Reeducation (Nr)

     b.    Frame conversion (Fc)

     c.    Grey scale and binary conversion(Scb)

**Step 03.**     **Human body land mark detection()**

        Fun(Hum_Det)

        Fun(Landmark_det)

        $T_{He}^{q} \leftarrow T_{He}^{q-1} + \Delta T_{He}^{q-1}$

        Fun(Human_2D_Stick_model)

        $Ubph = h \bowtie nq \bowtie S\_e \bowtie S\_hn$

        $\text{Mph} = S\_t \bowtie Ubph$

        $HLbs = hk \bowtie f \bowtie Mph$

        Fun(3D_Human_recons)

        $C_{me} = \overline{P_a}(e_x,e_y)\blacksquare P_{a+1}(e_x,e_y$

     a.    Human Detection (Hd)

     b.    Landmark detection(Ld)

     c.    Human 2D stick model(Hsm)

     d.    3D Human reconstruction(Hr)

**Step 04.**     **Features extraction()**

        Fun(Peri  Motion)

        $PM(t) = \alpha \sin(\omega t + k)$

        Fun(Non_Peri_Motion)

        $NPm(t) = \| \overline{P_{t,t+1}} - P_{t,t+2} \|$

        Fun(Motion_Direction_Flow)

        $M_{df} = \sum_{0}^{p} I_{vl}(I) \rightarrow D$

        Fun(Rotational_Angular_Joint)

        $A1 = \cos(x,y) \rightarrow L,\ A2 = \cos(x,y) \rightarrow L$

        $A3 = \cos(x,y) \rightarrow L,\ A4 = \cos(x,y) \rightarrow L$

     a.    PeriodicMotion(Pm)

     b.    NonPeriodicMotion(mn)

     c.    MotionDirectionandFlow(Mdf)

     d.    RotationalAngularJoint(Raj)

**Step 05.**     **Data optimization(Association based approach)**

        $nf_{(x,y)} = (g_x - g_y)(\dfrac{\Sigma_x - \Sigma_y}{2})(g_x - g_y)^t$

**Step 06.**     **Event classification()**

        CNN Based event classification approach ()

        $TR_p = \sum_{q} wi_{p.q} \times a_p + b_q$

**Step 07.**     **Output(Event Classification) Oc**

The graphical superstations show the flow of every program, system, or software. Figure 2 shows the graphical repetition in the shape of a flow chart of the proposed methodology.

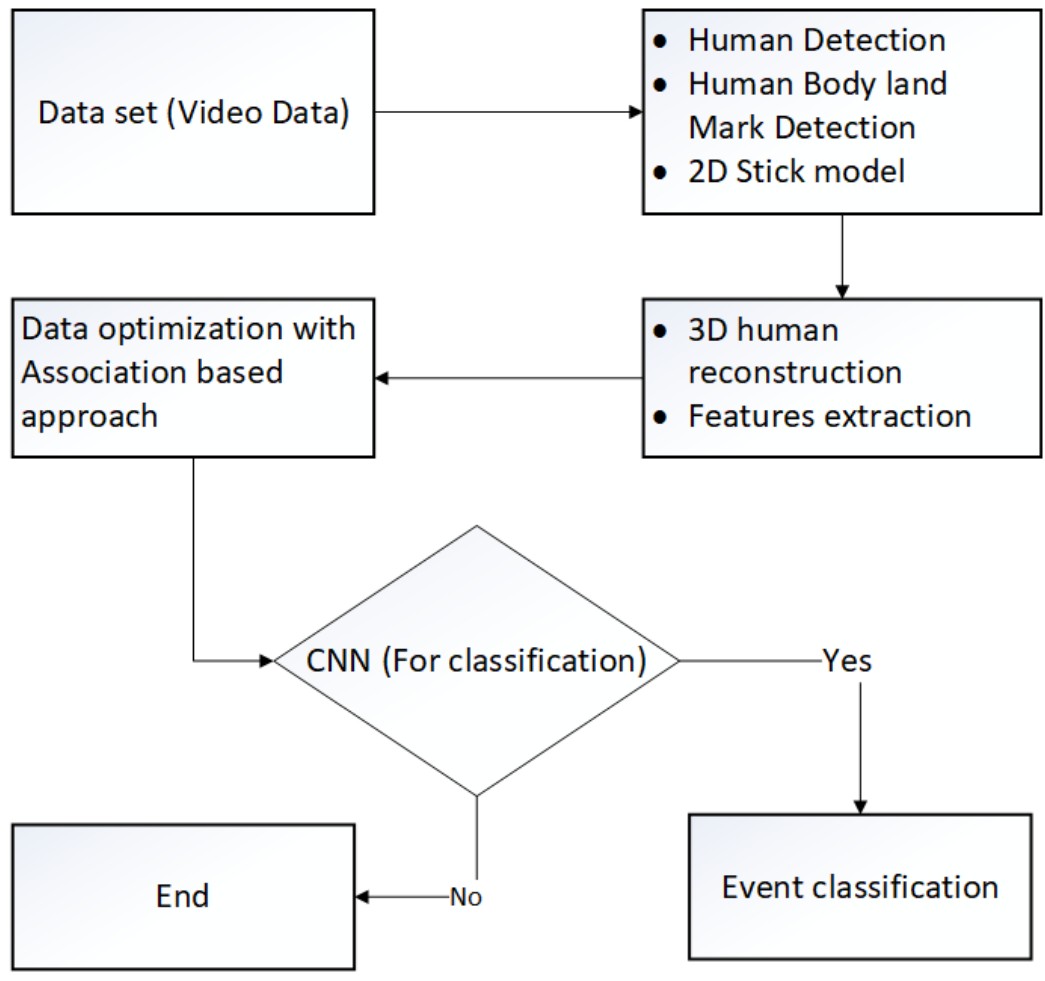

**Figure 2   Flow chart of the proposed method.**

## Preprocessing of the data

Before the detection of human body landmarks, some preprocessing methods are applied to save computational cost and time. Initially, video data is converted into images and then a motion blur filter is applied to reduce excess information.

## Background subtraction

For background subtraction, we applied an optimized merging method technique in which we initially applied Markov random field based on color information and region merging methods. After this, change detection in image sequence is applied over an adaptive threshold-based approach, floor detection, and finally spatial–temporal differencing is adopted to get more accurate results. Figure 3 shows the results of background subtraction techniques.

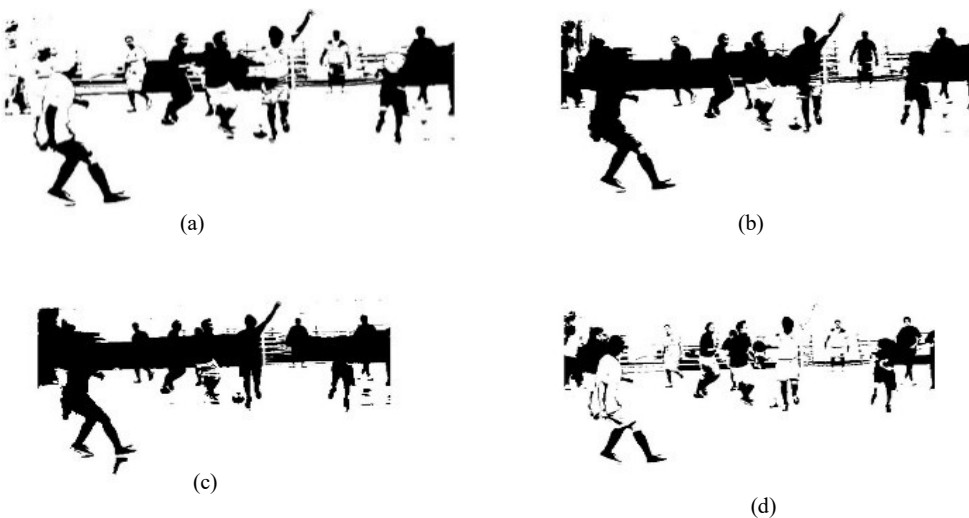

(a)

(b)

(c)

(d)

**Figure 3  Results of different background subtraction techniques.** (A) Change detection, (B) floor detection, (C) Markov random field and (D) spatial–temporal differencing.

## Silhouette optimization and human detection

In this sub-phase of landmark detection, we find the optimized human silhouette through the merging of change detection, floor detection, Markov random field, and spatial–temporal differencing techniques with the help of an adaptive threshold approach. (Algorithm 2) shows the detailed procedure of silhouette optimization.

---

**Algorithm 2: Human Silhouette Optimization**

**Input:** EHS: Extracted Human Silhouettes
**Output**: Optimized human silhouette
/* human body localization in input data*/
/* WP is for white area*/
/* OS is optimized human silhouette*/
/* SF is denoting shape feature*/

**Step 1:**
**Repeat**
      **For** k=1 to I **do**
            **For** k=1 to I **do**
                  $search$(WP)
            **End**
      **End**
      **If** WP1 > WP
            WP = WP1
      **End**
**Until** largest object shape searched in given frame.

**Step 2:**
/* Compare both $WP$ */
**For** all pixel in both $WP$
      **If** $WP_{pixel\ information\ of\ frame\ 1} = WP_{information\ of\ frame\ 2}$
            $WP_{pixel\ information\ of\ frame\ 3} = WP_{pixel\ information\ of\ frame\ 1}$
      **End**
      **If** $WP$ is inadequate for all inputs
            **If** pixel information is equal with SP
                  $OS = WP_{pixel}$
            **End**
      **End**
**End**

---

After this, human detection is performed in two phases, initially, head detection is performed with the help of a human head "size and shape-based" technique. We set the weight of a human head as w0 =1/25 of the human silhouette and, using region of interest model, we find the super pixel position of the human body and, after this, Gaussian kernel is used to capture the likely area of the human head. Finally, using this human head information, human shape and appearance information, human body movement, and motion information, human detection and identification is performed. Equation (1) is used for head tracking

$$T_{He}^q \leftarrow T_{He}^{q-1} + \Delta T_{He}^{q-1} \tag{1}$$

where $T_{He}^q$ represents a human head land-mark location in any given video frame $q$ which is consequential to calculating by the frame differences. For human detection, Eq. (2) shows the mathematical relationship.

$$T_{FH}^q = (T_{He}^q \leftarrow T_{He}^{q-1} + \Delta T_{He}^{q-1}) + T_{End}^q \tag{2}$$

where $T_{FH}^q$ represents a human location in any given video frame $q$ and $T_{End}^q$ shows the bounding box size for human detection. Figure 4 shows the results of optimized human body silhouettes, head detection, and human detection.

Once human silhouette extraction and human detection are achieved, the next phase is to find human body landmarks for the posture estimation and analysis of the human body movements.

## Body landmarks detection

In this sub-phase of landmark detection, we establish human body landmarks using a fast marching algorithm; we have applied this to the full human silhouette. Initially, the center point of the human body is extracted for the distance value $dis(h) = 0$, where $h$ is the initial point and is distinguished as a marked point. All remaining unmarked points of the human body are considered as $dis(p) = \infty$. this process is applied to every detected point and to the pixel value of the human silhouette. The mathematical representation is:

$$dis = \left\{ \frac{dis_x + dis_y + \sqrt{\Delta}}{2} \right\} when \ \Delta \geq 0 \tag{3}$$

$$\min(dis_x + dis_y) + wi \ otherwise \tag{4}$$

$$\Delta = x^2 - (dis_x - dis_y)^2$$

where $dis_x$ and $dis_y$ is the geodesic distance in the 2d plane, correspondingly, $dis_y = min(Disi + 1, mo, Disi - 1, mo)$ and $dis_y = min(Disi, no + 1, Di, no - 1)$. After this, human body parts estimation is performed by finding the midpoint of the human body and the hands, elbows, neck, head, knees, and feet points are extracted *Gochoo et al. (2021)*. The detection of the human midpoint is represented as;

$$T_t^q \leftarrow T_{to}^{q-1} + \Delta T_{to}^{q-1} \tag{5}$$

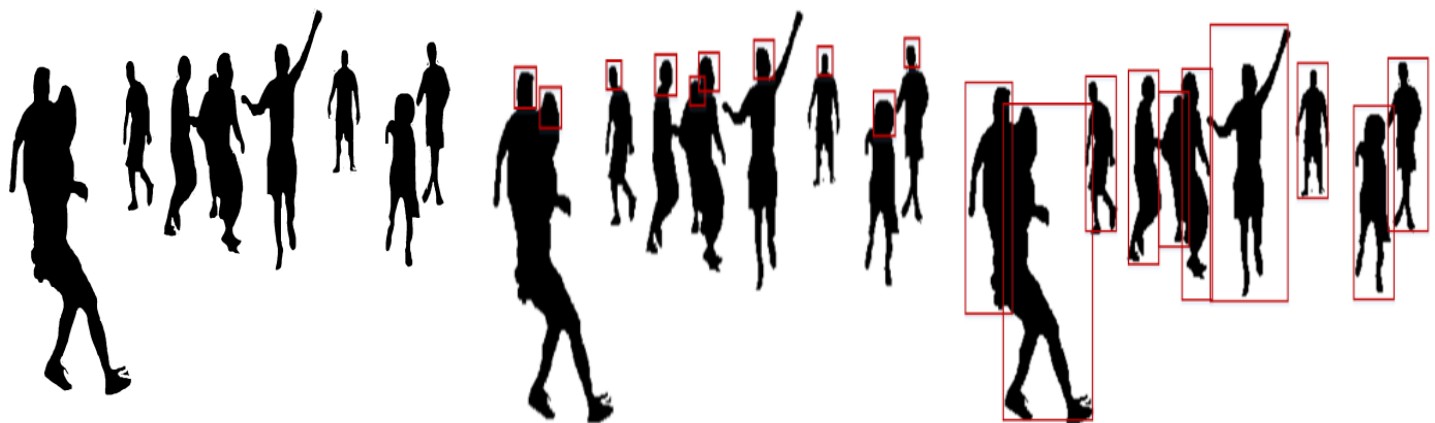

**Figure 4** Results of (A) optimized human silhouette (B) human head detection (C) human detection in RGB videos and image sequences.

where $T_t^q$ represents a human midpoint location in any given video frame q which is consequential to calculation according to frame differences *Akhter, Jalal & Kim (2021)*. To find the knees points we take the midpoint between the human-body midpoint and the two feet points. Equation (6) demonstrates the human knee points;

$$T_k^q = \left(T_t^q - T_f^q\right)/2 \tag{6}$$

where $T_k^q$ is a knee point, $T_m^q$ is the human body midpoint, and $T_f^q$ denotes a foot point. For each elbow position estimation we utilized the neck point and respective (left/right) hand point information and found the mid point between the hand and the neck points. Eq. (7)*as*;

$$T_e^q = (T_{hn}^q - T_{nq}^q)/2 \tag{7}$$

where $T_e^q$ denotes the human elbow point, $T_{hn}^q$ is the human hand point, and $T_{nq}^q$ denotes the neck point. Figure 5 represents the results of landmarks and body parts.

In this section, basic 2D human body skeletonization is achieved, using extracted human body key points. Figure 6 represents the detailed view of the human body 2D skeleton (*Jalal, Akhtar & Kim, 2020*; *Jalal, Khalid & Kim, 2020*) over eleven human body parts. The human body 2D skeleton is based on three main body areas: Upper body parts of the human body *Ubph*, Mid parts of the human body *Mph*, and lower body parts of the human body *Lbph.Ubph* denote the connection of the human head (*h*), neck point (*nq*), elbow (*e*), and human hand points (*hn*). *Mph* is initiated through the human midpoint (*t*) and *Unph.Lbph* is based on human knee points (*hk*) and footpoints (*f*). Every human body part or joint takes a specific letter k to complete a particular action. Equations (8), (9) and (10) demonstrate the calculated associations of the human 2D stick model as;

$$Ubph = h \bowtie nq \bowtie S_e \bowtie S_h n \tag{8}$$

$$Mph = S_t \bowtie Ubph \tag{9}$$

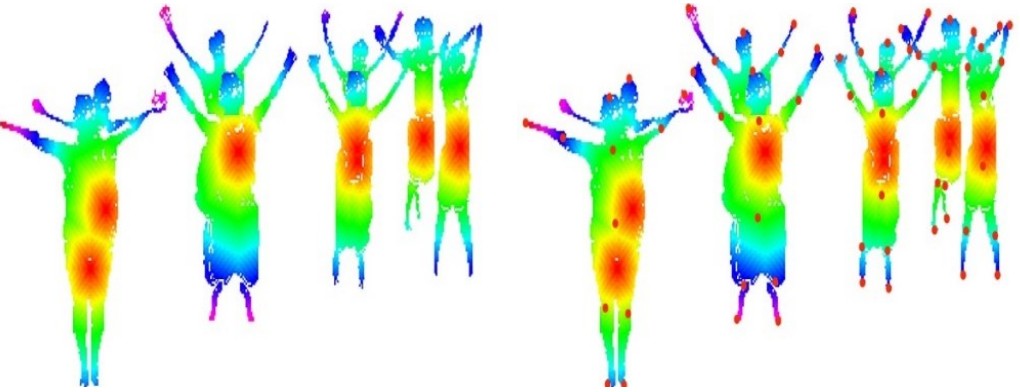

**Figure 5** Human body landmark detection results: (A) The landmark results using an HSV color map, (B) the eleven human body points.

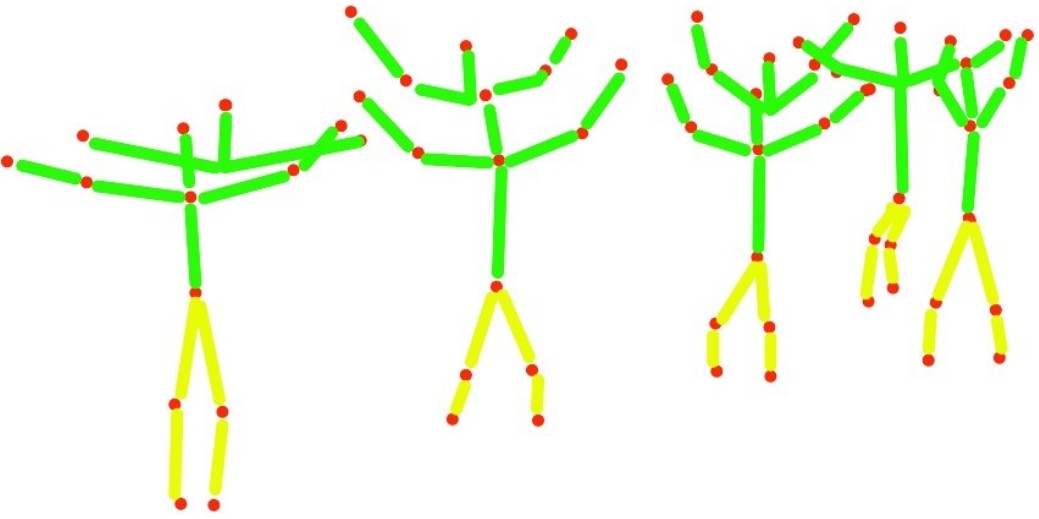

**Figure 6** The human 2D skeleton model results in over eleven human body parts.

$$HLbs = hk \bowtie f \bowtie Mph \qquad (10)$$

### 3D human reconstruction

For the reconstruction of the 3D human shape, we utilized a human 2D skeleton as a base step and as information for human posture estimation. For the reconstruction of the 3D human shape, we used the human body parts location and information.

### *A. Computational model with ellipsoids*

In this subpart of the 3D human reconstruction, the first step is achieved with a computational model with ellipsoid techniques. We take the joints information, the

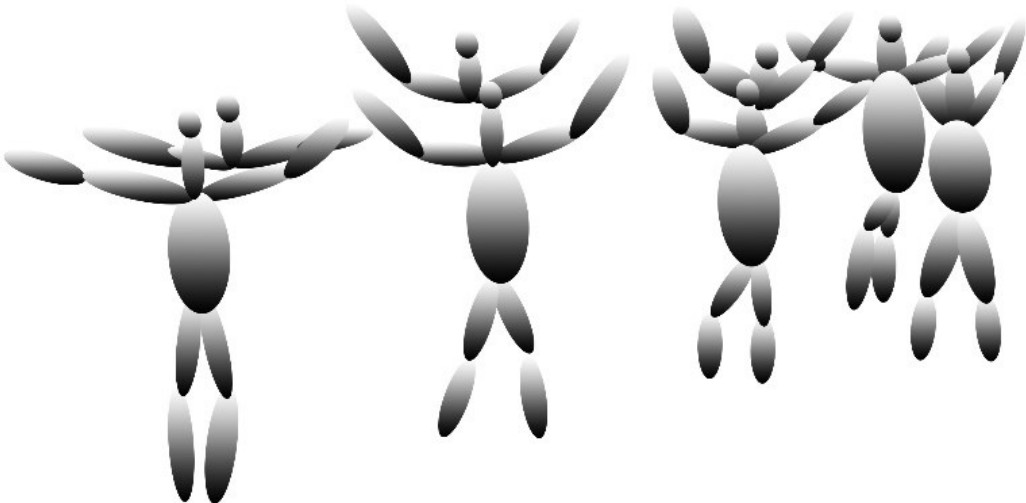

**Figure 7 The results of the computational model with ellipsoids over human body points.**

head point is connected with the neck point by an ellipsoid shape, and the neck point is connected with elbow points *via* an ellipsoid shape. The Elbow point is connected with the hand *via* an ellipsoid shape and the same procedure is followed for the remaining human body points.

$$C_{me} = P_a(e_x, e_y) \blacksquare P_{a+1}(e_x, e_y) \tag{11}$$

where $C_{me}$ is the computational model with ellipsoids, $P_a e_x, e_y e_z$ is the first point with the value of $x, y$ and $P_{a+1}(e_x, e_y)$ is the next point of the human body with the value of $x, y$. Figure 7 shows the results of the computational model with ellipsoids over human body points.

After completion of the computational model with ellipsoids, the next phase is the synthetic model with super quadrics.

### B. Synthetic model with super quadrics
To display the human posture and estimation of human motion the synthetic model with super quadrics is adopted, using the computational model with ellipsoids information we utilize the previous ellipsoids and convert them into a rectangular shape for more accurate information and analysis of human posture.

$$S_{SQ} = C_{me} \rightarrow s_e \tag{12}$$

where $S_{SQ}$ is the synthetic model with super quadrics and $C_{me}$ is the computational model with ellipsoid information, $\rightarrow s_e$ is the reshaping of the given information of points. Figure 8 shows the results of the synthetic model with super quadrics over the computational model with ellipsoids.

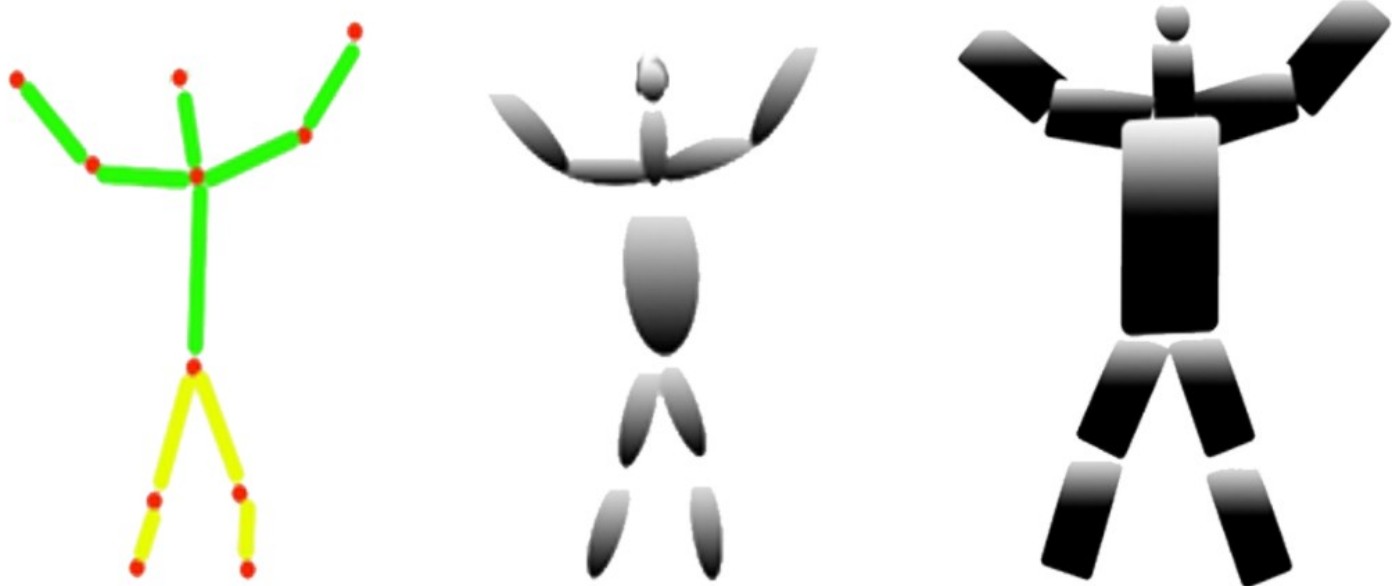

**Figure 8 The results of the synthetic model with super quadrics over human body points.** (A) Human 2D skeleton, (B) computational model with ellipsoids, (C) synthetic model with super quadrics.

### C. Joint angle estimation

For 3D reconstruction of the ellipsoids, the prerequisite step is to estimate joint angle information. For this, volumetric data and the edge information of the human body key points are extracted. For movement and angle information using the global and local coordinate system, we estimate the DOF for human body key points root information. After that, the Cartesian product of the skeleton graph is estimated for further processing.

$$T_{Ne}^{q} \leftarrow T_{Ne}^{q-1} + \Delta T_{Ne}^{q-1} \tag{13}$$

where $T_{Ne}^{q}$ represents the neck landmark location in any given video frame, $q$ is consequential to calculation of the frame differences. See Fig. 9.

$$F_v = \left[\theta_{g\_l}, \theta_{S\_h}, \theta_{S\_n}, \theta_{R\_e}, \theta_{L\_e}, \theta_{R\_h}, \theta_{L\_h}, \theta_{S\_m}, \theta_{R\_k}, \theta_{L\_k}, \theta_{R\_f}, \theta_{L\_f},\right] \tag{14}$$

where $F_v$ represents the angle joint function, $\theta_{g\_l}$ denotes the global to local coordinates, $\theta_{S\_h}$ indicates the head point, $\theta_{S\_n}$ shows the neck point, $\theta_{R\_e}$ denotes the right elbow point, $\theta_{L\_e}$ indicates the left elbow point, $\theta_{R\_h}$ represents the right-hand point, $\theta_{L\_h}$ shows the left-hand point, $\theta_{S\_m}$ indicates the mid-point, $\theta_{R\_k}$ shows the right knee point, $\theta_{L\_k}$ denotes the left knee point, $\theta_{R\_f}$ indicates the right foot point, $\theta_{L\_f}$ shows the left foot point.

### D. 3D ellipsoid reconstructions

Finally, the 3D reconstruction of the human body is implemented using human body joint information, ellipsoid information, and skeleton graphs. The preview of the 3D image gives us more precise and accurate posture information and estimation for further processing.

$$R_E(x, 0|I, D) \propto R_E(x)R_E(I|x)R_E(D|x)R_E(D|\theta) \tag{15}$$

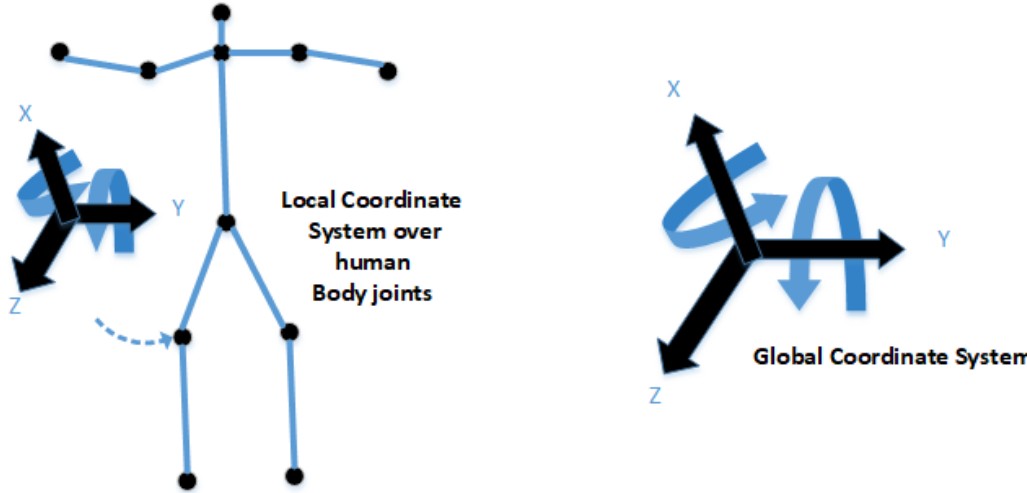

**Figure 9 The theme concept of local and global coordinate systems.** The left side shows the local coordinate system over the human left knee; the right side shows the DOF based global coordinate system.

where $R_E(x, 0|I, D)$ is the 3D reconstruction ellipsoid, $\propto$ is the reshaping and $R_E(x)R_E(I|x)R_E(D|x)R_E(D|\theta)$ shows the angle information based on previous data. Figure 10 shows the results of the 3D ellipsoid reconstruction over the synthetic model with super quadrics and joint angle estimation.

## Contextual features extraction

In this section, the extraction of contextual features is implemented in which DOF, periodic motion, non-periodic motion, motion direction and flow and rotational angular joint features are extracted.

### A. Degree of freedom

In contextual features extraction, Degree of freedom (DOF) is implemented over all body parts and the x,y,z dimension information In DOF features vector three directional angle values for each body parts, for knee points x_knee, y_knee, z_knee, for head points x_head, y_head, z_head, for neck points x_ neck, y_ neck, z_ neck, for elbow points x_ elbow, y_ elbow, z_ elbow, for hand points x_ hand, y_ hand, z_ hand, for midpoints x_ mid, y_ mid, z_ mid, for foot points x_ foot, y_ foot, z_ foot. Equation (16) shows the mathematical relation for DOF.

$$D_{of} = A(\theta x, \theta y, \theta z) \uparrow \varepsilon D \tag{16}$$

where $D_{of}$ represents the degree of freedom feature vector, $\theta x, \theta y, \theta z$ shows the three dimension of the angle and $\varepsilon D$ is the local and global coordinate system. Figure 11 shows the results for the degree of freedom:

### B. Periodic motion

In this contextual feature, human motion is detected over human body parts. The targeted area of interest is the human body portion which provides periodic motion. The detection

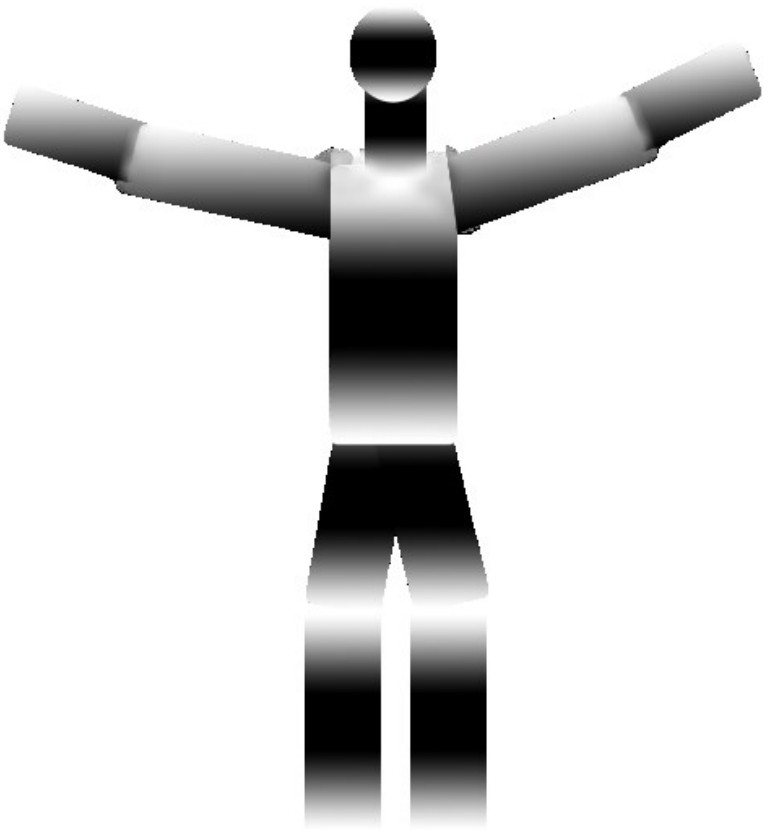

**Figure 10  The results of the 3D ellipsoid reconstruction over the synthetic model with super quadrics and joint angle estimation.**

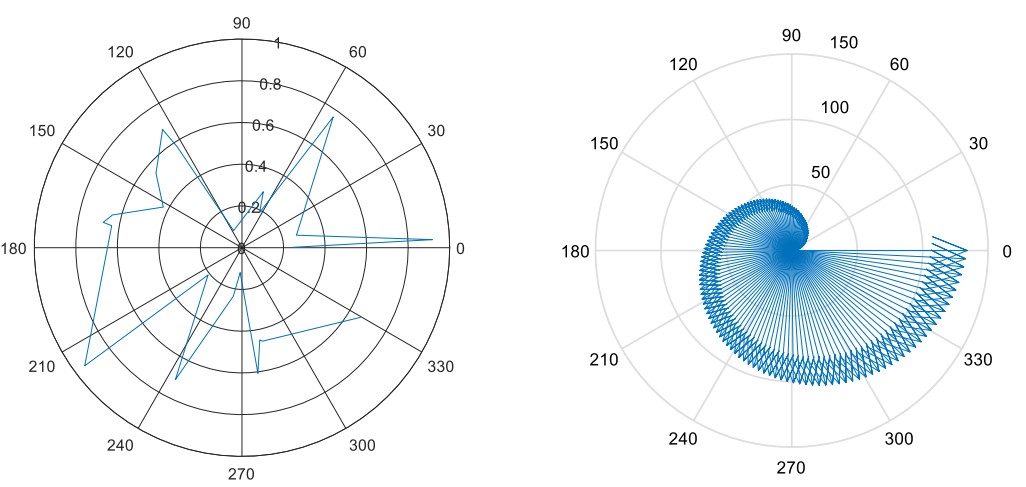

**Figure 11  Few DOF results examples.**

of this area is performed with the base analysis of the human body. A bounding box indicates the region of interest. The Eq. (17) shows the mathematical relation for periodic motion.

$$PM(t) = \alpha \sin(\omega t + k) \tag{17}$$

where $PM(t)$ denotes periodic motion and $\alpha \sin(\omega t + k)$ shows the relation of human motion that is repeated in any given sequence of images.

### C. Non-periodic motion

In the non-periodic motion contextual feature, human motion is detected over human body parts. The targeted area of interest is the human body portion which provides nonperiodic motion and non-uniform motion. The detection of this area is performed with the base analysis of human body motion. A bounding box indicates the region of interest. Equation (18) shows the mathematical relation of non-periodic motion:

$$NPm(t) = \| P_{t,t+1} - P_{t,t+2} \| \tag{18}$$

where $NPm(t)$ denotes non-periodic motion and $P_{t,t+1} - P_{t,t+2}$ shows the difference between the first and the next sequence of images in input data.

### D. Motion direction flow

For the identification of more accurate gait events motion direction flow is one a contributions in terms of contextual features. Using changes in motion and human motion body flow we detect the direction of human body movement and motion flow. Equation (19) shows the mathematical model for motion direction flow features.

$$M_{df} = \sum_{0}^{p} I_{vl}(I) \rightarrow D \tag{19}$$

where $M_{df}$ is motion direction flow of the human body, $I$ is the index values of the given image, $I_{vl}$ is RGB (x,y,z) pixel indexes, and $\rightarrow D$ shows the motion direction.

### E. Rotational angular joint

Rotational angular joint features are based on the angular geometry of human body parts. A 5 X 5 pixel region is used over detected body parts and from every node of the window of pixel region $\cos\theta$ is estimated and maps all values in the feature vector. The Eq. (20) shows the mathematical model of rotational angular joint features.

$$A1 = \cos(x,y) \longrightarrow L, A2 = \cos(x,y) \longrightarrow L \tag{20}$$

$$A3 = \cos(x,y) \longrightarrow L, A4 = \cos(x,y) \longrightarrow L$$

where $A1, A2, A3, A4$ denotes the sides of the $5 \times 5$ windows, $\cos(x,y)$ represents the angle value over pixel $x$ and $y$, and $\longrightarrow L$ indicates the side to follow. Figure 12 shows the results of rotational angular joint features over the dance class.

After the completion of the contextual features portion, we concatenate all the sub-feature vectors into the main feature vector while (Algorithm 3)

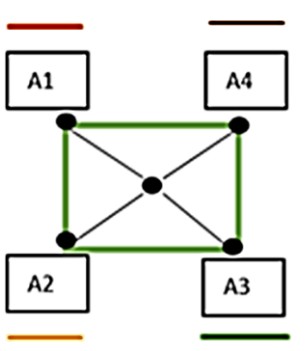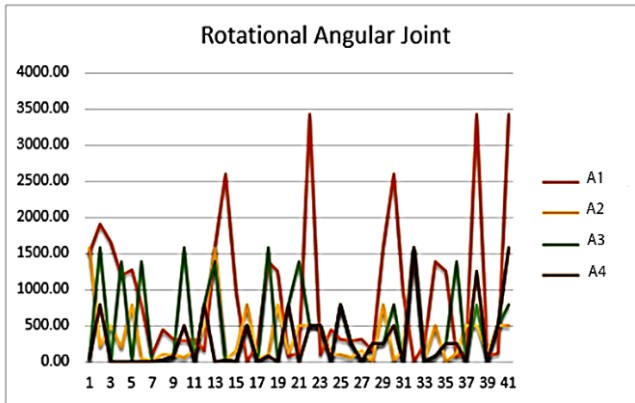

**Figure 12** Rotational angular joint results and the pattern of rotational angels.

shows the detailed overview of the contextual features extraction approach.

---

**Algorithm 3: Contextual Features Extraction**
Input: HS: Human silhouette from RGB video data
Output: Contextual feature vectors($Cf_1, Cf_2, Cf_3, ..., Cf_n$)
% feature vector for %
Contextual_features_vec ←[]
CF_vecsize ← GetFeaturesVectorsize ()
% loop over human silhouettes %
**For** i = 1:K
Contextual_features_vec _ interactions ← Get_ Contextual_features_vec (interactions)
    % extracting DOF, periodic motion, non periodic motion, motion direction and flow,
Rotational angular joint%
*DOF* ←ExtractDOF(Contextual_features_vec _ interactions)
*PeriodicMotion*←ExtractPeriodicMotion (Contextual_features_vec _ interactions)
*NonPeriodicMotion* ← ExtractNonPeriodicMotion (Contextual_features_vec _ interactions)
*MotionDirectionandFlow*←ExtractMotionDirectionandFlow(Contextual_features_vec_interactions)
*RotationalAngularJoint*←ExtractRotationalAngularJoint(Contextual_features_vec _ interactions)
*Contextualvectors* ← GetCFeaturevector
FVectors.append (CF_vectors)
**End**
*Contextualvectors* ← Normalize (Contextual_features_vec)
 **return** Contextual_features_vec ($Cf_1, Cf_2, Cf_3, ..., Cf_n$)

---

## Data optimization and features mining

The association rule-based features mining method helps us to pick the most unique features that screen out unnecessary and inconsistent features from the extracted dataset which tend to reduce gait event classification precision and accuracy. This is a bottom-up strategy that starts with a null feature set *nf* and progressively adds innovative features based on optimization function selection. This can decrease the mean square error which results in more significant details. The association rule-based features mining approach is

commonly used in various domains such as security systems, medical systems and image processing-based smart systems.

This technique helps to minimize the main features space data while the features mining approach is dependent upon the specific objective function for an optimal solution which plays the key role in gait event classification. The Bhattacharyya distance calculation features optimization approach is used in the present architecture for various human event-based classes. It can determine the differentiation rating $nf_{(x,y)}$ among different segments $x$ and $b$ and then test it.

$$nf_{(x,y)} = (g_x - g_y)(\frac{\Sigma_x - \Sigma_y}{2})(g_x - g_y)^t \tag{21}$$

where $nf_{(x,y)}$ is the optimal features set, $g_x$ are the mean and $\Sigma_x$ are the covariance of class $x$ and $g_y$ are the mean and $\Sigma_y$ are the covariance of class $y$ for $M$ numbers of event-based classes. The optimal solution score is computed as.

$$OFv = \frac{1}{N^2}\sum_{u=1}^{M}\sum_{v=1}^{M} nf_{(u,v)} \tag{22}$$

A recognition assessment criterion is suggested for estimating different gait event classifications for an input dataset to acquire selected features that can be expected to eliminate classification errors as well as provide improved inter-class interpretability throughout features data. For the mpii-video-pose dataset DOF, Periodic motion, Rotational angular joint features are selected. For the Pose_track dataset, DOF, Motion direction and flow nonPeriodic motion features are selected. For the COCO dataset, Motion direction and flow nonPeriodic motion, Rotational angular joints features are selected. Figure 13 shows the most accurate features results over the mpii-video pose, the COCO, and the Pose track datasets.

## Event classification

The extracted optimal features vector is used as input for Convolution Neural Network (CNN) for gait event classification. CNN is a deep learning-based classification approach which is widely used in image and video types of input data. CNN works well and it gives more accurate results compared to other traditional techniques. CNN adds less processing weight with minimum bias, thus providing a high accuracy rate.

The input, output, and hidden layers are the three main layers in a Convolutional Neural Network (CNN). The convolutional layer, important mechanisms, complete linked layer, and standardization layer are the four sub-divisions of each secret layer. The sub-band extracted features are across in the input neurons and highly correlated throughout the convolutional layer by a $5 \times 5$ graded selector. The batch normalization method is then used to aggregate the responses among all neuronal populations. To further minimize feature dimensions, clustered solutions are convolutional and combined again, and then the interaction maps are computed by the completely connected sheet.

$$TR_p = \sum_{q} wi_{p.q} \times a_p + b_q \tag{23}$$

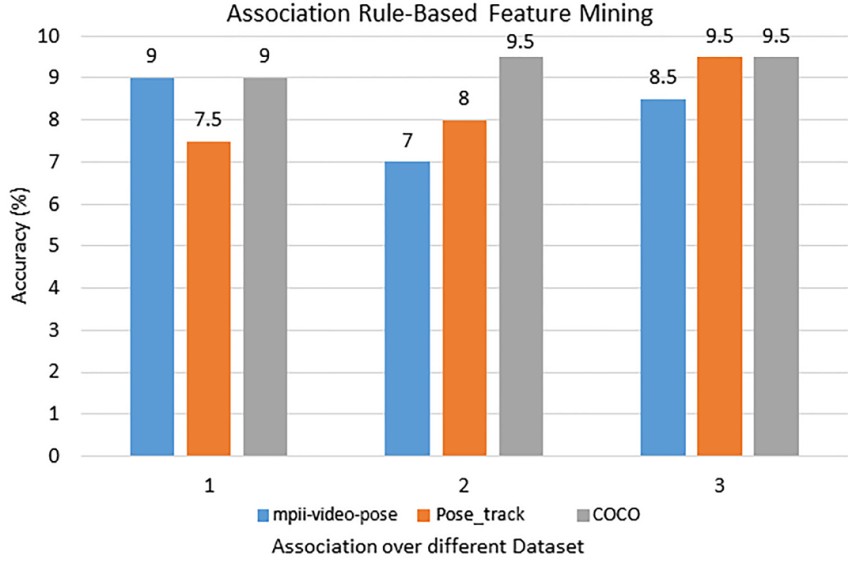

**Figure 13** The most accurate features results via the rule-based features mining approach over the mpii-video pose, COCO, and Pose track datasets.

where $TR_p$ is the CNN transfer function, $wi_{p.q}$ is the connecting layer's adjacent weight, $a_p$ shows the input optimized features vectors and $b_q$ is the bias values. Furthermore, the regression algorithm is adopted for parameter optimization and it reduces backpropagation errors. The output of the regression algorithm $\sigma(TR_p)$ returns the distribution function of total probability for possible $n$ repetition over the output layer of CNN.

$$\sigma(TR_p) = \frac{e^{TR_p}}{\sum_{n=1}^{n} e^{TR_p}}, q = 1, \ldots \ldots .n \tag{24}$$

Figure 14 shows the detailed process of CNN parameters learning over gait event detection and classification:

## RESULTS

### Dataset descriptions

The Mpii-video pose data set is a large-scale dataset, which contains human activities and posture information-based videos. 21 different activities such as home activities, lawn, garden, sports, washing windows, picking fruit, and rock climbing. All the videos All the videos selected for our dataset collection have been recommended as YouTube top 10 videos in each activity. Figure 18 shows some example images of the Mpii-video-pose dataset.

The COCO (Common Objects in Context) dataset is based on multi-person tracking and object detection dataset, different activities contains in the COCO dataset They include Bicycling, Conditioning exercise, Dancing, Fishing and hunting, Music playing, Religious activities, Sports, Transportation, Walking, Water activities, winter activities. Figure 19 shows some example images of the COCO dataset:
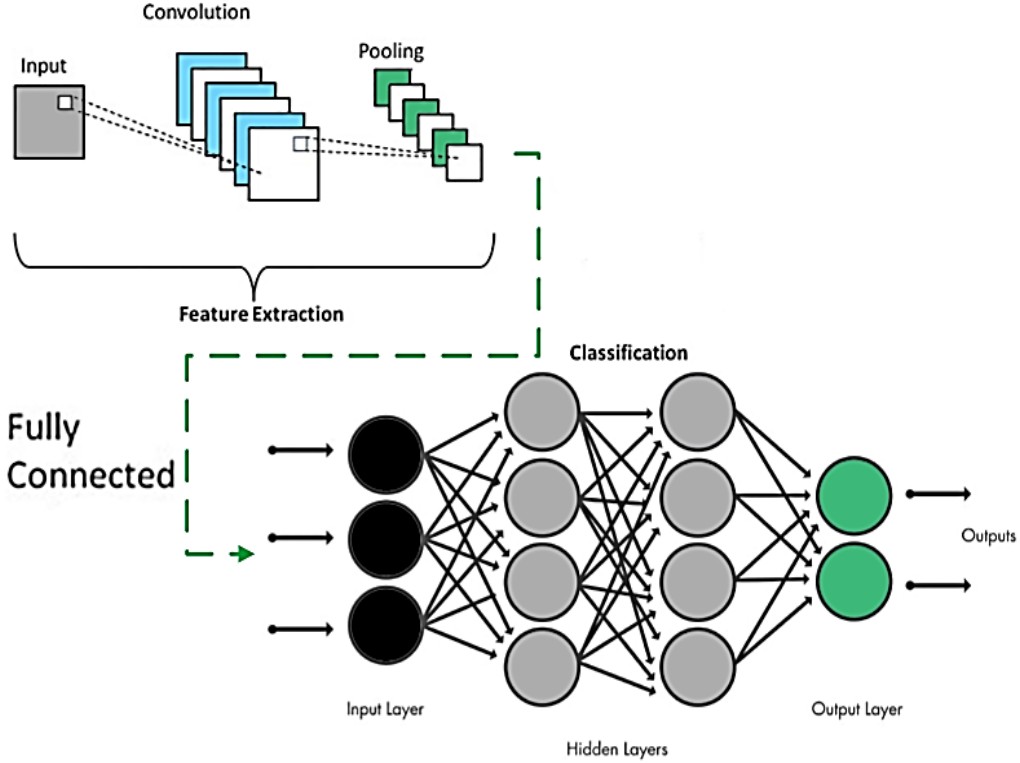

**Figure 14   CNN model overview.**

Pose track dataset is based on two main tasks including multi-person human pose estimation and analysis over a single frame. Videos and articulated tracking have been based on human posture estimation. Dataset mostly consists of complex videos such as crowded or team sports. Various activities have been covered in the pose track dataset. Figure 20 shows some example images of the pose track dataset.

## Experiment I: the landmarks detection accuracies

To calculate the effectiveness and precision of the detected body parts, we approximate the geodesic distance (*Akhter, 2020*; *Akhter, Jalal & Kim, 2021a*; *Akhter, Jalal & Kim, 2021b*) from the given ground truth (GT) of the input datasets by Eq. (25):

$$Die = \sqrt{\sum_{n=1}^{M}\left(\frac{O_n}{P_n} - \frac{O_n}{P_n}\right)^2} \tag{25}$$

Here, $O$ is the ground truth values of the datasets and $P$ is the current location of the recognized body part. The error margin of 16 is set to identify the accuracy among the acknowledged body part value and the input data. Through Eq. (26), the proportion of the recognized body parts encircled within the error margin value of the considered data is

**Table 2  Human body parts recognition and detection accuracy.**

| Body key points | Distance | MPII (%) | Distance | COCO (%) | Distance | Posetrack (%) |
|---|---|---|---|---|---|---|
| HP | 11.2 | 88 | 9.70 | 88 | 9.90 | 91 |
| NP | 10.8 | 86 | 10.2 | 86 | 11.1 | 88 |
| REP | 11.5 | 82 | 10.1 | 83 | 14.1 | 86 |
| RHP | 12.1 | 81 | 11.7 | 82 | 12.7 | 83 |
| LEP | 11.1 | 83 | 11.9 | 79 | 11.0 | 88 |
| LHP | 12.0 | 77 | 11.7 | 81 | 12.0 | 79 |
| MP | 10.1 | 91 | 13.1 | 90 | 11.9 | 91 |
| LKP | 13.2 | 94 | 12.8 | 92 | 12.3 | 87 |
| RKP | 9.90 | 91 | 10.3 | 91 | 11.7 | 81 |
| LFP | 10.3 | 94 | 11.2 | 95 | 14.1 | 94 |
| RFP | 11.5 | 91 | 10.3 | 94 | 13.8 | 97 |
| **Mean accuracy rate** | | **87.09** | | **87.36** | | **87.72** |

known as;

$$De = \frac{100}{n}\left[\sum_{n=1}^{i}\begin{cases}1 & if & De \leq 16 \\ 0 & if & De > 16\end{cases}\right] \tag{26}$$

In Table 2, columns 2, 4, and 6 show the error distances from the given dataset ground truth and columns 3, 5, and 7 show the body part recognition and detection accuracies over the MPII, COCO and Posetrack datasets respectively.

HP = Head point, NP = Neck point, REP = Right elbow point, RHP = Right hand point, LEP = Left elbow point, LHP = Left hand point, MP = Mid-point, LKP = left knee point, RKP = Right knee point, LFP = Left foot point, RFP = Right foot point.

Table 3 represents the results of multi-person human body parts for the mpii-video-pose dataset. For identified body parts, we indicate with ✓ and for unidentified we adopted we use ×. We attained a detection accuracy for human1 -63.63%, human2 - 72.72%, human3 −63.63%, human4- 72.72%, human5 - 72.72% and the mean detection accuracy of 69.09%.

HP = Head point, NP = Neck point, REP = Right elbow point, RHP = Right hand point, LEP = Left elbow point, LHP = Left hand point, MP = Mid-point, LKP = left knee point, RKP = Right knee point, LFP = Left foot point, RFP = Right foot point.

Table 4 represents the results of multi-person human body parts for the COCO dataset. For identified body parts, we indicated ✓ and for unidentified body parts we adopted ×. We a attained detection accuracy for human1 -81.81%, human2 - 72.72%, human3 - 72.72%, human4- 72.72%, human5 - 72.72%, and the mean detection accuracy of 74.54%.

HP = Head point, NP = Neck point, REP = Right elbow point, RHP = Right hand point, LEP = Left elbow point, LHP = Left hand point, MP = Mid-point, LKP = left knee point, RKP = Right knee point, LFP = Left foot point, RFP = Right foot point.

Table 5 presents the results of multi-person human body parts for the pose track dataset. For identified body parts, we use ✓ and for unidentified we adopted use ×. The detection accuracies follow: for human1- 63.63%, human2 - 63.63%, human3 - 63.63%, human4 - 63.63%, human5 - 72.72% and the mean detection accuracy is 65.45%.

**Table 3  Human body parts results of multi-person for mpii-video-pose dataset.**

| Body parts | Human1 | Human2 | Human3 | Human4 | Human5 |
|---|---|---|---|---|---|
| HP | ✓ | ✓ | ✓ | ✓ | ✗ |
| NP | ✗ | ✓ | ✗ | ✗ | ✓ |
| REP | ✓ | ✓ | ✓ | ✓ | ✓ |
| RHP | ✓ | ✗ | ✗ | ✓ | ✓ |
| LEP | ✗ | ✓ | ✓ | ✓ | ✗ |
| LHP | ✗ | ✓ | ✓ | ✗ | ✓ |
| MP | ✓ | ✗ | ✓ | ✓ | ✓ |
| LKP | ✓ | ✓ | ✗ | ✗ | ✓ |
| RKP | ✗ | ✗ | ✓ | ✓ | ✗ |
| LFP | ✓ | ✓ | ✗ | ✓ | ✓ |
| RFP | ✓ | ✓ | ✓ | ✓ | ✓ |
| Accuracy | 63.63% | 72.72% | 63.63% | 72.72% | 72.72% |
| **Mean accuracy = 69.09%** | | | | | |

**Table 4  Human body parts results of multi-person for COCO dataset.**

| Body parts | Human1 | Human2 | Human3 | Human4 | Human5 |
|---|---|---|---|---|---|
| HP | ✓ | ✓ | ✓ | ✓ | ✓ |
| NP | ✓ | ✓ | ✓ | ✗ | ✓ |
| REP | ✓ | ✗ | ✗ | ✓ | ✓ |
| RHP | ✓ | ✗ | ✗ | ✗ | ✗ |
| LEP | ✓ | ✓ | ✓ | ✓ | ✓ |
| LHP | ✗ | ✓ | ✓ | ✓ | ✗ |
| MP | ✗ | ✗ | ✓ | ✓ | ✓ |
| LKP | ✓ | ✓ | ✗ | ✗ | ✓ |
| RKP | ✓ | ✓ | ✓ | ✓ | ✗ |
| LFP | ✓ | ✓ | ✓ | ✓ | ✓ |
| RFP | ✓ | ✓ | ✓ | ✓ | ✓ |
| Accuracy | 81.81% | 72.72% | 72.72% | 72.72% | 72.72% |
| **Mean accuracy = 74.54%** | | | | | |

HP = Head point, NP = Neck point, REP = Right elbow point, RHP = Right hand point, LEP = Left elbow point, LHP = Left hand point, MP = Mid-point, LKP = left knee point, RKP = Right knee point, LFP = Left foot point, RFP = Right foot point.

## Experiment II: event classification accuracies

For gait event classification, we used a CNN-based deep learning approach. The design method is evaluated by the Leave One Subject Out (LOSO) cross-validation method. In Fig. 15, the results over the mpii-video-pose dataset show 90.90% gait event classification and detection accuracy. After this, we applied the deep belief network over the Olympic sports dataset and found the stochastic remote sensing event classification results. Figure 16 represents the confusion matrix for the COCO dataset with 89.09% mean accuracy for gait event classification. Finally, CNN is applied over the pose track dataset, with the mean gait

**Table 5   Human body parts results of multi-person for mpii-video-pose dataset.**

| Body parts | Human1 | Human2 | Human3 | Human4 | Human5 |
|---|---|---|---|---|---|
| HP | ✓ | ✓ | ✓ | ✓ | ✓ |
| NP | ✗ | ✗ | ✓ | ✗ | ✓ |
| REP | ✓ | ✓ | ✗ | ✓ | ✓ |
| RHP | ✗ | ✗ | ✓ | ✗ | ✗ |
| LEP | ✓ | ✓ | ✗ | ✓ | ✓ |
| LHP | ✓ | ✗ | ✓ | ✗ | ✗ |
| MP | ✗ | ✓ | ✓ | ✓ | ✓ |
| LKP | ✓ | ✓ | ✗ | ✓ | ✓ |
| RKP | ✓ | ✗ | ✓ | ✗ | ✗ |
| LFP | ✓ | ✓ | ✗ | ✓ | ✓ |
| RFP | ✗ | ✓ | ✓ | ✓ | ✓ |
| Accuracy | 63.63% | 63.63% | 63.63% | 63.63% | 72.72.1% |
| Mean accuracy = 65.45% | | | | | |

|  | Bi | Ce | Da | Fh | Mp | Ra | Sp | Tr | Wl | Wa | Wn |
|---|---|---|---|---|---|---|---|---|---|---|---|
| Bi | 9 | 1 | 0 | 0 | 0 | 0 | 0 | 0 | 0 | 0 | 0 |
| Ce | 0 | 9 | 0 | 0 | 0 | 0 | 1 | 0 | 0 | 0 | 0 |
| Da | 0 | 0 | 10 | 0 | 0 | 0 | 0 | 0 | 0 | 0 | 1 |
| Fh | 0 | 1 | 0 | 9 | 0 | 0 | 0 | 0 | 0 | 0 | 0 |
| MP | 1 | 0 | 0 | 0 | 9 | 0 | 0 | 0 | 0 | 0 | 0 |
| Ra | 0 | 0 | 1 | 0 | 0 | 9 | 0 | 0 | 0 | 0 | 0 |
| SP | 0 | 1 | 0 | 0 | 0 | 0 | 9 | 0 | 0 | 0 | 0 |
| Tr | 1 | 0 | 0 | 0 | 0 | 0 | 0 | 8 | 0 | 0 | 1 |
| Wl | 0 | 1 | 0 | 0 | 0 | 0 | 0 | 0 | 9 | 0 | 1 |
| Wa | 0 | 0 | 0 | 0 | 0 | 0 | 0 | 0 | 0 | 10 | 0 |
| Wn | 0 | 0 | 0 | 0 | 1 | 0 | 0 | 0 | 0 | 0 | 9 |
| **Gait event detection and classification mean accuracy = 90.90%** | | | | | | | | | | | |

**Figure 15   Confusion matrix results using CNN over Mpii-video-pose dataset.**

event classification accuracy of 88.18%. Figure 17 represents the results in the shape of the confusion matrix for the pose track dataset.

BI = Bicycling, Ce = Conditioning exercise, Da = Dancing, Fh = Fishing and hunting, Mp = Music playing, Ra = Religious activities, SP = Sports, Tr = Transportation, Wi = Walking, Wa = Water activities, Wn = Winter activities.

BI = Bicycling, Da = Dancing, Ce = Conditioning exercise, Fh = Fishing and hunting, Ra = Religious activities, Mp = Music playing, SP = Sports, Wi = Walking, Tr = Transportation, Wa = Water activities, Wn = Winter activities.

Ce = Conditioning exercise, BI = Bicycling, Da = Dancing, Mp = Music playing, Fh = Fishing and hunting, Ra = Religious activities, Tr = Transportation, SP = Sports, Wi = Walking, Wa = Water activities, Wn = Winter activities.

|     | Bi | Da | Ce | Fh | Ra | Mp | Sp | Wl | Tr | Wa | Wn |
| --- | --- | --- | --- | --- | --- | --- | --- | --- | --- | --- | --- |
| Bi  | 8  | 0  | 0  | 0  | 1  | 0  | 0  | 0  | 1  | 0  | 0  |
| Da  | 0  | 10 | 0  | 0  | 0  | 0  | 0  | 0  | 0  | 0  | 0  |
| Ce  | 0  | 0  | 9  | 0  | 1  | 0  | 0  | 0  | 0  | 0  | 0  |
| Fh  | 0  | 0  | 0  | 8  | 1  | 1  | 0  | 0  | 0  | 0  | 0  |
| Ra  | 0  | 0  | 0  | 0  | 10 | 0  | 0  | 0  | 0  | 0  | 0  |
| Mp  | 0  | 0  | 1  | 0  | 0  | 8  | 0  | 0  | 1  | 0  | 0  |
| Sp  | 1  | 0  | 0  | 0  | 0  | 0  | 9  | 0  | 0  | 0  | 0  |
| Wl  | 0  | 0  | 0  | 0  | 1  | 0  | 0  | 9  | 0  | 0  | 0  |
| Tr  | 0  | 1  | 0  | 0  | 0  | 0  | 0  | 0  | 8  | 0  | 1  |
| Wa  | 0  | 0  | 1  | 0  | 0  | 0  | 0  | 0  | 0  | 9  | 0  |
| Wn  | 0  | 0  | 0  | 0  | 0  | 0  | 0  | 0  | 0  | 0  | 10 |

**Gait event detection and classification mean accuracy = 89.09%**

**Figure 16   Confusion matrix results using CNN over the COCO dataset.**

|     | Ce | Bi | Da | Mp | Fh | Ra | Tr | Sp | Wl | Wn | Wa |
| --- | --- | --- | --- | --- | --- | --- | --- | --- | --- | --- | --- |
| Ce  | 7  | 0  | 0  | 1  | 0  | 0  | 1  | 0  | 1  | 0  | 0  |
| Bi  | 0  | 8  | 1  | 0  | 0  | 0  | 0  | 0  | 0  | 1  | 0  |
| Da  | 0  | 0  | 9  | 0  | 0  | 0  | 0  | 0  | 1  | 0  | 0  |
| Mp  | 0  | 0  | 0  | 10 | 0  | 0  | 0  | 0  | 0  | 0  | 0  |
| Fh  | 0  | 0  | 0  | 0  | 10 | 0  | 0  | 0  | 0  | 0  | 0  |
| Ra  | 0  | 0  | 0  | 0  | 0  | 10 | 0  | 0  | 0  | 0  | 0  |
| Tr  | 0  | 0  | 0  | 0  | 1  | 0  | 8  | 0  | 0  | 1  | 0  |
| Sp  | 0  | 0  | 0  | 1  | 0  | 0  | 0  | 9  | 0  | 0  | 0  |
| Wl  | 0  | 0  | 0  | 0  | 0  | 0  | 0  | 0  | 10 | 0  | 0  |
| Wn  | 0  | 0  | 0  | 0  | 1  | 1  | 0  | 0  | 0  | 8  | 0  |
| Wa  | 0  | 0  | 0  | 0  | 1  | 0  | 0  | 0  | 0  | 1  | 8  |

**Gait event detection and classification mean accuracy = 88.18%**

**Figure 17   Confusion matrix results using CNN over the Pose track dataset.**

## Experiment III: comparison with other classification algorithms

In this segment, we equate the recall, precision, and f-1 measure over the mpii-video-pose dataset, the COCO, and the posetrack dataset. For the classification of gait events we used Decision tree, Artificial Neural Network and we associated the consequences with the CNN. (Table 6) shows the results over the mpii-video-pose dataset, (Table 7) shows the results over the COCO dataset, and (Table 8) shows the results over the posetrack dataset.

BI = Bicycling, Ce = Conditioning exercise, Da = Dancing, Fh = Fishing and hunting, Mp = Music playing, Ra = Religious activities, SP = Sports, Tr = Transportation, Wi = Walking, Wa = Water activities, Wn = Winter activities.

BI = Bicycling, Da = Dancing, Ce = Conditioning exercise, Fh = Fishing and hunting, Ra = Religious activities, Mp = Music playing, SP = Sports, Wi = Walking, Tr = Transportation, Wa = Water activities, Wn = Winter activities.

**Table 6  Precision, recall, and F-1 measure comparison with the artificial neural network, decision tree and CNN over Mpii-video-pose dataset.**

| Event classes | Artificial neural network | | | Decision tree | | | CNN | | |
|---|---|---|---|---|---|---|---|---|---|
| Events | Precision | Recall | F-1 measure | Precision | Recall | F-1 measure | Precision | Recall | F-1 Measure |
| Bi | 0.778 | 0.700 | 0.737 | 0.667 | 0.600 | 0.632 | 0.818 | 0.900 | 0.857 |
| Ce | 0.700 | 0.700 | 0.700 | 0.700 | 0.700 | 0.700 | 0.692 | 0.900 | 0.783 |
| Da | 0.857 | 0.600 | 0.706 | 0.818 | 0.900 | 0.857 | 0.909 | 0.909 | 0.909 |
| Fh | 0.909 | 1.000 | 0.952 | 0.727 | 0.800 | 0.762 | 1.000 | 0.900 | 0.947 |
| MP | 0.900 | 0.900 | 0.900 | 0.727 | 0.800 | 0.762 | 0.900 | 0.900 | 0.900 |
| Ra | 0.889 | 0.800 | 0.842 | 0.889 | 0.800 | 0.842 | 1.000 | 0.900 | 0.947 |
| SP | 0.727 | 0.889 | 0.800 | 0.833 | 1.000 | 0.909 | 0.900 | 0.900 | 0.900 |
| Tr | 0.875 | 0.778 | 0.824 | 0.909 | 1.000 | 0.952 | 1.000 | 0.800 | 0.889 |
| Wl | 0.875 | 0.700 | 0.778 | 1.000 | 0.700 | 0.824 | 1.000 | 0.818 | 0.900 |
| Wa | 0.818 | 1.000 | 0.900 | 1.000 | 0.900 | 0.947 | 1.000 | 1.000 | 1.000 |
| Wn | 0.769 | 1.000 | 0.870 | 0.800 | 0.800 | 0.800 | 0.750 | 0.900 | 0.818 |
| **Mean** | **0.827** | **0.824** | **0.819** | **0.825** | **0.818** | **0.817** | **0.906** | **0.893** | **0.896** |

**Table 7  Precision, recall, and F-1 measure comparison with the artificial neural network, decision tree and CNN over COCO dataset.**

| Event classes | Artificial neural network | | | Decision tree | | | CNN | | |
|---|---|---|---|---|---|---|---|---|---|
| Events | Precision | Recall | F-1 measure | Precision | Recall | F-1 measure | Precision | Recall | F-1 Measure |
| Bi | 0.818 | 0.750 | 0.783 | 0.700 | 0.700 | 0.700 | 0.889 | 0.800 | 0.842 |
| Da | 0.750 | 0.750 | 0.750 | 0.889 | 0.800 | 0.842 | 0.909 | 1.000 | 0.952 |
| Ce | 0.889 | 0.667 | 0.762 | 0.833 | 1.000 | 0.909 | 0.818 | 0.900 | 0.857 |
| Fh | 0.900 | 1.000 | 0.947 | 0.818 | 0.900 | 0.857 | 1.000 | 0.800 | 0.889 |
| Ra | 0.909 | 0.909 | 0.909 | 0.818 | 0.900 | 0.857 | 0.714 | 1.000 | 0.833 |
| Mp | 0.900 | 0.818 | 0.857 | 0.900 | 0.900 | 0.900 | 0.889 | 0.800 | 0.842 |
| Sp | 0.667 | 0.857 | 0.750 | 0.889 | 0.800 | 0.842 | 1.000 | 0.900 | 0.947 |
| Wl | 0.900 | 0.818 | 0.857 | 0.727 | 1.000 | 0.842 | 1.000 | 0.900 | 0.947 |
| Tr | 0.900 | 0.750 | 0.818 | 0.889 | 0.800 | 0.842 | 0.800 | 0.800 | 0.800 |
| Wa | 0.800 | 1.000 | 0.889 | 1.000 | 0.800 | 0.889 | 1.000 | 0.900 | 0.947 |
| Wn | 0.727 | 1.000 | 0.842 | 0.875 | 0.700 | 0.778 | 0.909 | 1.000 | 0.952 |
| Mean | 0.833 | 0.847 | 0.833 | 0.849 | 0.845 | 0.842 | 0.903 | 0.891 | 0.892 |

Ce = Conditioning exercise, BI = Bicycling, Da = Dancing, Mp = Music playing, Fh = Fishing and hunting, Ra = Religious activities, Tr = Transportation, SP = Sports, Wi = Walking, Wa = Water activities, Wn = Winter activities.

## Experiment IV: performance analysis and comprehensive analysis of features selection

For gait event classification, contextual features were proposed in this article. Four features are based upon the full human body that are DOF, periodic motion, non-periodic motion, and motion direction flow. One is based upon human body joints which are the rotational angular joints. Using these features we make a complete features vector to classify gait event

**Table 8 Precision, recall, and F-1 measure comparison with the artificial neural network, decision tree and CNN over Posetrack dataset.**

| Event classes | Artificial neural network | | | Decision tree | | | CNN | | |
|---|---|---|---|---|---|---|---|---|---|
| Events | Precision | Recall | F- 1 measure | Precision | Recall | F- 1 measure | Precision | Recall | F- 1 Measure |
| Ce | 0.818 | 0.750 | 0.783 | 0.769 | 1.000 | 0.870 | 1.000 | 0.700 | 0.824 |
| Bi | 0.700 | 0.700 | 0.700 | 0.750 | 0.818 | 0.783 | 1.000 | 0.800 | 0.889 |
| Da | 0.889 | 0.667 | 0.762 | 0.875 | 0.700 | 0.778 | 0.900 | 0.900 | 0.900 |
| Mp | 0.900 | 1.000 | 0.947 | 0.778 | 0.700 | 0.737 | 0.833 | 1.000 | 0.909 |
| Fh | 0.875 | 0.875 | 0.875 | 0.692 | 0.900 | 0.783 | 0.769 | 1.000 | 0.870 |
| Ra | 0.900 | 0.818 | 0.857 | 0.818 | 0.900 | 0.857 | 0.909 | 1.000 | 0.952 |
| Tr | 0.750 | 0.900 | 0.818 | 0.700 | 0.700 | 0.700 | 0.889 | 0.800 | 0.842 |
| Sp | 0.857 | 0.750 | 0.800 | 0.875 | 0.700 | 0.778 | 1.000 | 0.900 | 0.947 |
| Wl | 0.857 | 0.667 | 0.750 | 1.000 | 0.750 | 0.857 | 0.833 | 1.000 | 0.909 |
| Wn | 0.800 | 1.000 | 0.889 | 1.000 | 1.000 | 1.000 | 0.727 | 0.800 | 0.762 |
| Wa | 0.769 | 1.000 | 0.870 | 0.778 | 0.778 | 0.778 | 1.000 | 0.800 | 0.889 |
| Mean | 0.829 | 0.830 | 0.823 | 0.821 | 0.813 | 0.811 | 0.896 | 0.882 | 0.881 |

detection with the help of data mining technique and CNN-based classification approach. To check the performance of our contextual features we adopted one more experiment in which various combinations of features are utilized to check the best combination. Initially, the combination of degree of freedom (DOF), periodic motion, and nonperiodic motion are used. After that, the combination of degree of freedom (DOF), periodic motion, nonperiodic motion, and motion direction flow, finally the combination of degree of freedom (DOF), periodic motion, nonperiodic motion, motion direction flow, and the rotational angular joint are used. Table 9 shows the overview of performance analysis. The complete combination shows more accurate results.

DOF = degree of freedom, PM = periodic motion, NM = nonperiodic motion, MDF = motion direction flow, RAJ = rotational angular joint

## Experiment V: comparison of our proposed system with state-of-the-art techniques

*Fan et al. (2015)* developed a unique approach to estimate the human pose which is based on deep learning-based dual-source CNN. As the input they used patches of a given image and human body patches. After that they combined both contextual and local index values to estimate human posture with a better accuracy rate. *Pishchulin et al. (2016)* proposed a robust formulation as a challenge of subsection partitioning and labeling (SPLP). The SPLP structure, unlike previous two-stage methods that separated the identification and pose estimation measures, suggests the number of persons, certain poses, spatial proximity, including component level occlusions all at the same time. In *Wei et al. (2016)*, convolutional position devices have an edge infrastructure for solving formal classifications based on computer vision that does not require visual type reasoning. We demonstrated that by transmitting increasingly refined confusion beliefs among points, a sequential framework consisting of convolutional networks is incapable of effectively training a structural component for the position. *Jin et al. (2019)* proposed SpatialNet

**Table 9** Performance analysis over various features extracting upon the MPII, COCO and Pose track datasets.

| Features combination | MPII (%) | COCO (%) | Pose Track (%) |
|---|---|---|---|
| *DOF, PM, NPM* | 71.20 | 73.21 | 72.59 |
| *DOF, PM, NPM,MDF* | 76.73 | 76.07 | 75.13 |
| ***DOF, PM, NPM,MDF,RAJ*** | **90.90** | **89.09** | **88.18** |

**Table 10** Gait event mean accuracy comparison with the other methods over the MPII, COCO and Pose track datasets.

| Methods | MPII (%) | Methods | COCO (%) | Methods | Pose Track (%) |
|---|---|---|---|---|---|
| *Fan et al. (2015)* | 73.00 | *Sun et al. (2018)* | 74.20 | *Jin et al. (2019)* | 71.08 |
| *Pishchulin et al. (2016)* | 87.10 | *Rachmadi, Uchimura & Koutaki (2016)* | 82.30 | Bao et al. (2020) | 72.03 |
| *Wei et al. (2016)* | 90.50 | *Zhu et al. (2019)* | 83.10 | *Umer et al. (2020)* | 74.02 |
| **Proposed method** | **90.90** | | **89.09** | | **88.18** |

and TemporalNet combined to form a single pose prediction and monitoring conceptual model: Body part identification and part-level temporal classification are handled by SpatialNet, while the contextual classification of human events is handled by TemporalNet. *Bao et al. (2020)* suggest a hand gesture identification-by-tracking system that incorporates pose input into both the video human identification and human connection levels. A person's position prediction with pose descriptive statistics is used in the first level to reduce the impact of distracting and incomplete human identification in images. *Umer et al. (2020)* present a method for detecting people in video which depends on key feature connections. Rather than training the system to estimate key-point communications on video sequences, the system is equipped to estimate human pose utilizing personality on massive scale datasets. *Sun et al. (2018)* suggested a technique for features extracted in which they remove guided optical flow and use a CNN-based paradigm to identify and classify human events. *Rachmadi, Uchimura & Koutaki (2016)* described a method for dealing with event recognition and prevention using CNN and NNA (Network in Network Architecture) frameworks, which are the foundation of modern CNN. CNN's streamlined infrastructure, median, average, and commodity features are used to define human activities. *Zhu et al. (2019)* provide a detailed method for identifying incidents in security video. Throughout the TRECVID-SED 2016 test, their method outperformed others by a substantial margin by combining path modeling with deep learning. Table 10 shows the gait event mean accuracy comparison with the other methods over the MPII, COCO and Pose track datasets.

## SCOPE OF THE PAPER

This article is based on human motion analysis and gait event detection-based approaches, which are parts of computer vision, image processing, data science, machine learning, deep learning, neural networks, and artificial intelligence. Especially we can deploy this research project over airport security, railways station, seaports, bus stations, metro stations, real-time smart system environment and other smart surveillance systems. With

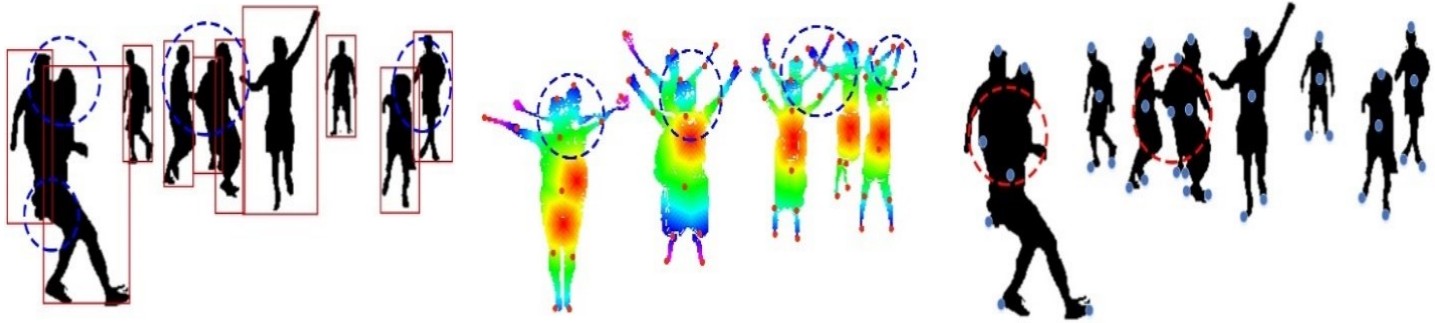

**Figure 18** Some examples of limitations and failure cases.

complex human-based crowed data which is the input of the proposed system, human motion analysis and gait event classification are the challenging tasks, with the help of various techniques, features, and classification method. We achieve this with a much better accuracy rate than previous methods. This system works fine in less crowed based datasets and also on complex video data systems, but sometimes accuracy may be compromised due to various factors such as distance, location, shadow, and other illusion-based factors. This system also works on the real-time smart system to analyze human motion and gait event detection.

## FAILURE CASES AND LIMITATIONS

This article is based on human motion analysis and gait event detection-based approaches, which are parts of computer vision, image processing, data science, machine learning, deep learning, neural networks, and artificial intelligence. Especially we can deploy this research project over airport security, railways station, seaports, bus stations, metro stations, real-time smart system environment and other smart surveillance systems. With complex human-based crowed data which is the input of the proposed system, human motion analysis and gait event classification are the challenging tasks, with the help of various techniques, features, and classification method. We achieve this with a much better accuracy rate than previous methods. This system works fine in less crowed based datasets and also on complex video data systems, but sometimes accuracy may be compromised due to various factors such as distance, location, shadow, and other illusion-based factors. This system also works on the real-time smart system to analyze human motion and gait event detection. Figure 18 shows the detailed overview of failures and limitations of the proposed method in which we can see the overlapping issue and complex scenarios to find human and human landmarks.

Openpose *Viswakumar et al. (2019)* is a CNN-based approach that is used for the human motion estimation and the gait analysis in various domains. We applied openpsoe CNN classification and detected the human body parts. The mean accuracy for the MPII dataset was 84.90%, the one for COCO dataset was 86.00% and the one for Posetrack dataset was 85.18%. Table 11 shows the complete results of openpose CNN.

**Table 11  Human body parts recognition and detection accuracy using Openpose CNN.**

| Body key points | Distance | MPII (%) | Distance | COCO (%) | Distance | Posetrack (%) |
|---|---|---|---|---|---|---|
| HP | 12.5 | 86 | 10.1 | 84 | 11.1 | 83 |
| NP | 12.3 | 83 | 12.3 | 83 | 12.5 | 81 |
| REP | 10.6 | 81 | 12.5 | 82 | 12.9 | 80 |
| RHP | 13.1 | 88 | 14.1 | 86 | 10.1 | 85 |
| LEP | 12.3 | 84 | 10.6 | 80 | 10.9 | 82 |
| LHP | 14.0 | 78 | 10.8 | 83 | 11.6 | 77 |
| MP | 11.3 | 88 | 12.2 | 88 | 9.9 | 92 |
| LKP | 11.0 | 85 | 11.9 | 87 | 10.5 | 89 |
| RKP | 15.1 | 83 | 13.3 | 89 | 12.8 | 85 |
| LFP | 14.2 | 90 | 11.6 | 91 | 11.3 | 91 |
| RFP | 12.3 | 88 | 11.1 | 93 | 10.1 | 92 |
| **Mean accuracy rate** | | **84.90** | | **86.00** | | **85.18** |

**Table 12  Comparison table of the Openpose CNN model with the proposed method.**

| Dataset | Openpose CNN | Proposed method |
|---|---|---|
| MPII (%) | 84.90 | 87.09 |
| COCO (%) | 86.00 | 87.36 |
| Posetrack (%) | 85.18 | 87.72 |

HP = Head point, NP = Neck point, REP = Right elbow point, RHP = Right hand point, LEP = Left elbow point, LHP = Left hand point, MP = Mid-point, LKP = left knee point, RKP = Right knee point, LFP = Left foot point, RFP = Right foot point.

Table 12 shows the detailed comparison of the openpose CNN model with the proposed method. Results show the better accuracy of our proposed method than the accuracy of the openpose method.

Figure 19 shows the detailed results of openpose CNN over human sillhouttes.

## CONCLUSION

This article is based on a reconstituted 3D synthetic model of the human body, gait event detection and classification over complex human video articulated datasets. Three benchmark datasets were selected for experiments: mpii-video-pose, COCO, and pose tracking datasets. Initially, human detection and landmark recognition are performed. After that, 2D human skeletons are transformed into 3D synthetic-based models for the analysis of human posture. For features reduction and optimization, a rule-based features mining technique is adopted and finally, a deep learning classification algorithm CNN is applied for gait event recognition and classification. For the mpii-video pose dataset, we achieve the human landmark detection mean accuracy of 87.09% and gait event recognition mean accuracy of 90.90%. For the COCO dataset, we achieve the human landmark detection mean accuracy of 87.36% and gait event recognition mean accuracy of 89.09%. For the pose track dataset, we achieve the human landmark detection mean accuracy of 87.72%

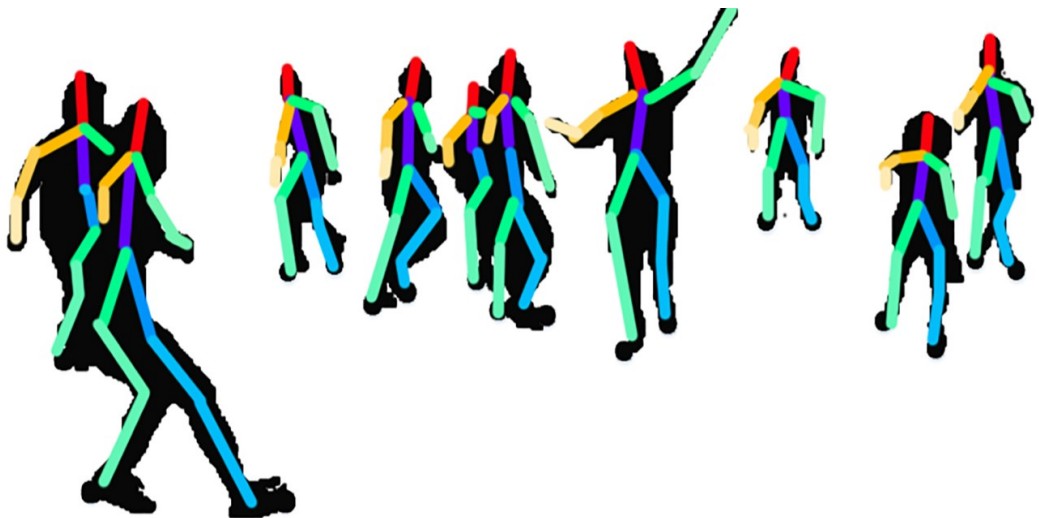

**Figure 19** Results of the openpose CNN based classification.

and gait event recognition mean accuracy of 88.18%. The proposed system's performance shows a significant improvement compared to existing state-of-the-art frameworks. The limitation of the proposed framework is due to the complexity in the videos and group density which make it is difficult to achieve more accurate results.

### Funding

This research was supported by the Basic Science Research Program through the National Research Foundation of Korea (NRF), funded by the Ministry of Education (No. 2018R1D1A1A02085645). Also, this work was supported by the Korea Medical Device Development Fund grant funded by the Korea government (the Ministry of Science and ICT, the Ministry of Trade, Industry and Energy, the Ministry of Health & Welfare, the Ministry of Food and Drug Safety) (Project Number: 202012D05-02). The funders had no role in study design, data collection and analysis, decision to publish, or preparation of the manuscript.

### Grant Disclosures

The following grant information was disclosed by the authors:
The Basic Science Research Program through the National Research Foundation of Korea (NRF), funded by the Ministry of Education: No. 2018R1D1A1A02085645.
The Korea Medical Device Development Fund grant funded by the Korea government (the Ministry of Science and ICT, the Ministry of Trade, Industry and Energy, the Ministry of Health & Welfare, the Ministry of Food and Drug Safety) (Project Number: 202012D05-02).

## Competing Interests

The authors declare there are no competing interests.

## Author Contributions

- Yazeed Ghadi performed the experiments, analyzed the data, authored or reviewed drafts of the paper, and approved the final draft.
- Israr Akhter conceived and designed the experiments, performed the experiments, performed the computation work, prepared figures and/or tables, and approved the final draft.
- Mohammed Alarfaj performed the computation work, authored or reviewed drafts of the paper, and approved the final draft.
- Ahmad Jalal performed the experiments, prepared figures and/or tables, and approved the final draft.
- Kibum Kim conceived and designed the experiments, analyzed the data, authored or reviewed drafts of the paper, and approved the final draft.

## Data Availability

The codes are available in the Supplementary Files. The third-party data are from:

1. COCO: https://cocodataset.org/#download
2. mpii-video-pose: http://human-pose.mpi-inf.mpg.de/#download
3. Pose track: https://posetrack.net/users/download.php

## Supplemental Information

Supplemental information for this article can be found online at http://dx.doi.org/10.7717/peerj-cs.764#supplemental-information.

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
