# Peer review of "Syntactic model-based human body 3D reconstruction and event classification via association based features mining and deep learning"

_PeerJ Computer Science, doi:10.7717/peerj-cs.764_

## Round 0.1 · original submission · Major Revisions

Three consistent reviews have been received. The reviewers pointed out some merits and places to be improved. Please provide a detailed one to one response.

·

Basic reporting

In this paper, authors presented a robust approach to human posture analysis and gait event detection from complex video-based data. For this, initially posture information, landmark information are extracted, and human 2D skeleton mesh are extracted, using this information set we reconstruct the human 2D to 3D model. Contextual features, namely, degrees of freedom over detected body parts, joint angle information, periodic and non-periodic motion, and human motion direction flow, are extracted. For features mining, authors applied the rule-based features mining technique and, for gait event detection and classification, the deep learning-based CNN technique is applied over the mpii-video pose, the COCO, and the pose track datasets

Experimental design

Experimental Design is Ok. But some more technical revisions are required in this paper with regard to Algorithm and Flowchart of the proposed methodology.

Validity of the findings

Data and Analysis experimentation based finiding are Ok.

Additional comments

The Paper needs the following Major Revisions:

1. Introduction should be more broad with regard to Problem Definition and even the scope of the paper. More information needs to be discussed in the paper.

2. Related works- In every related works, the stress should be on the novelty proposed, the explanation of the technique and even on experimental conclusions. At the end of related works, highlight in 9-15 lines what overall technical gaps are observed that led to the design of the proposed methodology.

3. Add the Algorithm, Flowchart and Steps of working of the proposed methodology.

4. Add some performance analysis section to this paper.

5. Add future scope to this paper.

Reviewer 2 ·

Basic reporting

Why not compare all methods on three data sets in Table 9?

Experimental design

no comment

Validity of the findings

no comment

Additional comments

1 According to the performance scores given in the tables, the results of the proposed method are compelling. But from a technical perspective, it is mainly a combination of existing technologies, and the contributions are not obvious. I hope that the manuscript will highlight the contributions of this paper in the revised version, omitting unnecessary formulas.
2 Please discuss the failure cases and the limitations of the proposed method.
3 Why not compare all methods on three data sets in Table 9?
4 “To find the knees points we take the midpoint between the human-body midpoint and the two feet points. Eq. (6) demonstrates the human knee points;” According to this manuscript at lines 204-206, the denominator in formula (1) is 2?
5 My personal suggestion is to intersperse the Figures in the manuscript.
6 Some paragraphs are indented too much (such as the first paragraph), and some are not indented.

·

Basic reporting

1. The writing in this article lacks proper structure, has various grammatical mistakes, and somethings are not clearly formulated.
2. The mathematical notation used in the manuscript is not easy to parse and doesn't follow standard notations, which makes it difficult for users to understand the ideas in detail. This also results in various redundancies throughout the manuscript.
I have marked some of them in the paper and attached them. Authors should make the manuscript easy for readers to follow.
3. Limitations and future works are not discussed.
4. All the figures and tables are added towards the end of the manuscript and are not placed correctly. Captions for the figures are not self-explanatory.
5. Some variables are not defined before they are used (see the attachment).
6. Relevant missing work: Openpose, Cao et. al.
7. Feature generation steps have various ambiguities and require clarification(see the attachment).
8. Algorithms should be re-written and some of them(like Algorithm 2) are not even required. Indention is not followed in algorithms

Experimental design

1. Goal and idea of the method is interesting. Authors propose various features like pose, joint angle, silhouette, etc, and use these for gait-event classification. Although all the features used seem important for gait classification, there can be some redundant features and some might enable wrong correlation. I would be very interested to see a comprehensive analysis of feature selection.
2. Since the method section is not clear, even though the techniques used in the method are trivial methods, I fear, paper cannot be exactly reproduced. This is because some things were not clear.
3. I have marked specific questions in the manuscript.

Validity of the findings

1. Datasets used in the paper are publicly available datasets and authors provide detailed evaluation for various types of motion.

2. Limitations and future works should be discussed in brief. Authors do not explain the limitations of their method like multi-step data preprocessing step, time etc.

3. Authors use CNN for classification, although CNN is powerful features extractors, how do author justify their choice of not using CNN based features? A comparative study or justification is required.

4. How is this pipeline of even gait classification better than OpenPose based 2d joint detection + gait classification layer. This is the very basic experimental baseline that comes to my mind, given OpenPose performs very well for most of the datasets and generalizes well.

Additional comments

I have corrected some grammatical mistakes and anomalies, not all. So you should carefully read the manuscript again along the same line.

---

## Round 0.2 · Minor Revisions

Please make the edits suggested by reviewers.

·

Basic reporting

In this paper, authors presented a robust approach to human posture analysis and gait event detection from complex video-based data. For this, initially posture information, landmark information are extracted, and human 2D skeleton mesh are extracted, using this information set we reconstruct the human 2D to 3D model. Contextual features, namely, degrees of freedom over detected body parts, joint angle information, periodic and non-periodic motion, and human motion direction flow, are extracted. For features mining, authors applied the rule-based features mining technique and, for gait event detection and classification, the deep learning-based CNN technique is applied over the mpii-video pose, the COCO, and the pose track datasets.

Experimental design

Experimental design is Ok and falls under aim and scope of the paper.

Validity of the findings

Yes, the revised findings are Ok.

Additional comments

The Paper stands Accepted with no further rrevisions.

·

Basic reporting

Following the suggestions and requests of all reviewers, the authors have updated the manuscript. The problem statement, technical gaps, experiments, algorithm, and flow charts have been updated/added. This improves the initial submission by a great margin.
The paper reads well now.

Experimental design

no comment

Validity of the findings

My previous concern of using CNN not just for classification but also for features has not been answered well. Also, authors do now validate their method against the openpose+CNN classifier baseline.

They only provide some argument, which doesn't seem strong enough. I would recommend the authors add openpose+CNN classifier baseline results in the paper, even if their method performs worse than this. It will be useful for readers to draw a conclusion.

Additional comments

My only request is that the authors add the above baseline(section 3) in the paper.

---

## Round 0.3 · accepted · Accept

I went through the revision and the responses. Reviewers' questions have been sufficiently addressed. I am happy to recommend acceptance without sending out for further review.